# The E3 ubiquitin ligase RNF220 maintains hindbrain *Hox* expression patterns through regulation of WDR5 stability

**Huishan Wang**[1†], **Xingyan Liu**[2,3†], **Yamin Liu**[1,4], **Chencheng Yang**[1,4], **Yaxin Ye**[1,4], **Xiaomei Yu**[1,5], **Nengyin Sheng**[1,6,7], **Shihua Zhang**[2,6,8*], **Bingyu Mao**[1,5,6*], **Pengcheng Ma**[1*]

[1]Key Laboratory of Genetic Evolution and Animal Models, Kunming Institute of Zoology, Chinese Academy of Sciences, Kunming, China; [2]Academy of Mathematics and Systems Science, Chinese Academy of Science, Beijing, China; [3]School of Mathematical Sciences, University of Chinese Academy of Sciences, Beijing, China; [4]Kunming College of Life Science, University of Chinese Academy of Sciences, Kunming, China; [5]National Resource Center for Non-Human Primates, Kunming Primate Research Center and National Research Facility for Phenotypic & Genetic Analysis of Model Animals (Primate Facility), Kunming Institute of Zoology, Chinese Academy of Sciences, Kunming, China; [6]Center for Excellence in Animal Evolution and Genetics, Chinese Academy of Sciences, Kunming, China; [7]Key Laboratory of Animal Models and Human Disease Mechanisms of Yunnan Province, Kunming Institute of Zoology, Chinese Academy of Sciences, Kunming, China; [8]Key Laboratory of Systems Biology, Hangzhou Institute for Advanced Study, University of Chinese Academy of Sciences, Chinese Academy of Sciences, Hangzhou, China

**\*For correspondence:**
zsh@amss.ac.cn (SZ);
mao@mail.kiz.ac.cn (BM);
kunmapch@mail.kiz.ac.cn (PM)

[†]These authors contributed equally to this work

## eLife Assessment

This **valuable** study focuses on gene regulatory mechanisms essential for hindbrain development. Through molecular genetics and biochemistry, the authors propose a new mechanism for the control of Hox genes, which encode highly conserved transcription factors essential for hindbrain development. The strength of evidence is **solid**, as most claims are supported by the data. This work will be of interest to developmental biologists.

**Abstract** The spatial and temporal linear expression of *Hox* genes establishes a regional *Hox* code, which is crucial for the antero-posterior (A-P) patterning, segmentation, and neuronal circuit development of the hindbrain. RNF220, an E3 ubiquitin ligase, is widely involved in neural development via targeting of multiple substrates. Here, we found that the expression of *Hox* genes in the pons was markedly up-regulated at the late developmental stage (post-embryonic day E15.5) in *Rnf220⁻/⁻* and *Rnf220⁺/⁻* mouse embryos. Single-nucleus RNA sequencing (RNA-seq) analysis revealed different *Hox* de-repression profiles in different groups of neurons, including the pontine nuclei (PN). The *Hox* pattern was disrupted and the neural circuits were affected in the PN of *Rnf220⁺/⁻* mice. We showed that this phenomenon was mediated by WDR5, a key component of the TrxG complex, which can be polyubiquitinated and degraded by RNF220. Intrauterine injection of WDR5 inhibitor (WDR5-IN-4) and genetic ablation of *Wdr5* in *Rnf220⁺/⁻* mice largely recovered the de-repressed *Hox* expression pattern in the hindbrain. In P19 embryonal carcinoma cells, the retinoic acid-induced *Hox* expression was further stimulated by *Rnf220* knockdown, which can also be rescued by *Wdr5*

knockdown. In short, our data suggest a new role of RNF220/WDR5 in *Hox* pattern maintenance and pons development in mice.

---

## Introduction

The coordinated expression of *Hox* genes clusters is critical for axial patterning to determine the positional identities during the development of many tissues, including the hindbrain (*Frank and Sela-Donenfeld, 2019*; *Parker et al., 2014*; *Parker and Krumlauf, 2020*; *Pöpperl and Featherstone, 1993*). In mammals, there are 39 *Hox* genes organized in four clusters (*Hoxa*, *Hoxb*, *Hoxc*, and *Hoxd*) and divided into 1–13 paralog groups (PG). During vertebrate development, the hindbrain forms eight metameric segmented units along the antero-posterior (A-P) axis known as rhombomeres (r), which give rise to the cerebellum, pons, and medulla later on *Ghosh and Sagerström, 2018*; *Parker and Krumlauf, 2020*. In the embryonic hindbrain, *Hox*1-5 genes are expressed in a nested overlapping pattern with rhombomere-specific boundaries. This *Hox* code is critical for the establishment of rhombomeric territories and the determination of cell fates (*Barsh et al., 2017*; *Parker et al., 2016*; *Parker and Krumlauf, 2020*).

At later embryonic stages, *Hox* genes also play vital roles in neuronal migration and circuit formation (*Feng et al., 2021*; *Hockman et al., 2019*; *Kratochwil et al., 2017*; *Wang et al., 2013*). For example, neurons in the pontine nuclei (PN), which acts as an information integration station between the cortex and cerebellum (*Kratochwil et al., 2017*; *Maheshwari et al., 2020*), originate at the posterior rhombic lip of r6-r8, migrate tangentially to the rostral ventral hindbrain, and then receive different inputs from the cortex, in which the specific expression pattern of *Hox3-5* genes determines the migration routines, cellular organization, and neuronal circuits (*Geisen et al., 2008*; *Kratochwil et al., 2017*; *Maheshwari et al., 2020*; *Di Meglio et al., 2013*; *Lizen et al., 2017*). Note that *Hox* gene expression is maintained up to postnatal and even adult stages in the hindbrain derivatives (*Farago et al., 2006*; *Feng et al., 2021*; *Kratochwil et al., 2017*).

*Hox* pattern maintenance is generally governed by epigenetic regulators, especially the polycomb-group (PcG) and trithorax-group (trxG) complexes, which repress and activate *Hox* genes, respectively (*Bahrampour et al., 2019*; *Chopra et al., 2009*; *Kang et al., 2022*; *Papp and Müller, 2006*). In mammals, different SET/MLL proteins and a common multi-subunit core module consisting of WDR5, RBBP5, ASH2L, and DPY-30 (WRAD) constitute the COMPASS (complex of proteins associated with Set1) family of TrxG complexes, which acts as a methyltransferase generally (*Cenik and Shilatifard, 2021*; *Jambhekar et al., 2019*; *Schuettengruber et al., 2017*). Among them, WDR5 plays a central scaffolding role and has been shown to act as a key regulator of *Hox* maintenance (*Wang et al., 2011*; *Wysocka et al., 2005*). WDR5 was also reported for *HOX* maintenance in human and associated with various blood and solid tumors. In acute myeloid leukemia and acute lymphoblastic leukemia, WDR5 is responsible for MLL-mediated *HOXA9* activation and is a potential drug target (*Chen et al., 2021*; *Yu et al., 2021*). In addition, WDR5 facilitates *HOTTIP*, a long noncoding RNA, to recruit the MLL complex and thus stimulate *HOXA9* and *HOXA13* expression during the progression of prostate and liver tumors (*Fu et al., 2017*; *Malek et al., 2017*; *Quagliata et al., 2014*; *Wong et al., 2020*).

The E3 ubiquitin ligase RNF220 is widely involved in neural development through targeting a range of proteins (*Kim et al., 2018*; *Kong et al., 2010*; *Ma et al., 2021*; *Ma et al., 2022a*; *Wang et al., 2022Wang et al., 2022*; *Ma et al., 2022c*). Although our previous reports have shown that *Rnf220⁻/⁻* mice is neonatal lethal and the survived *Rnf220⁺/⁻* mice develop severe motor impairments, the underlying mechanisms remain incompletely understood (*Ma et al., 2021*; *Ma et al., 2019*). In the present study, a markedly up-regulation of *Hox* genes is observed in the pons of both *Rnf220⁺/⁻* and *Rnf220⁻/⁻* mice during late embryonic development. In consequence, the topographic input connectivity of the PN neurons with cortex is disturbed in *Rnf220⁺/⁻* mice. Furthermore, we identified WDR5 as a direct target of RNF220 for K48-linked ubiquitination and thus degradation, a mechanism critical for *Hox* de-repression by *Rnf220* knockout. Together, these findings reveal a novel role for RNF220 in the regulation of *Hox* genes and pons development in mice.

## Results

### *Rnf220* insufficiency leads to dysregulation of *Hox* expression in late embryonic pons

*Rnf220* is strongly expressed in the embryonic mouse mid/hindbrain (*Ma et al., 2019*; *Wang et al., 2022*). Based on microarray analysis, we explored changes in expression profiles in E18.5 *Rnf220*[+/-] and *Rnf220*[-/-] mouse brains, which breed floxed *Rnf220* to *Vasa-Cre* mice. Interestingly, several *Hox* genes were up-regulated in the mouse brain of both genotypes (*Supplementary file 1*, *Supplementary file 2*). Considering that *Hox* expression in forebrain is too low (data not shown), we focus on the hindbrain and found that this phenomenon was observed only in late embryonic stages after E15.5 (*Figure 1—figure supplement 1*). Since *Rnf220*[-/-] mice is neonatal lethal and there is not significant difference in the upregulation of *Hox* genes between *Rnf220*[+/-] and *Rnf220*[-/-] mice, *Rnf220*[+/-] mice were used in subsequent assays in this study. To determine the specific brain regions with aberrant *Hox* expression, we examined the expression levels of *Hoxa9* and *Hoxb9*, two of the most significantly up-regulated genes, in different brain and spinal regions (*Figure 1—figure supplement 2A–D*). Notably, both *Hoxa9* and *Hoxb9* were exclusively up-regulated in the brainstem of *Rnf220*[+/-] mice (*Figure 1—figure supplement 2C and D*). The brainstem comprises the midbrain, pons, and medulla (*Figure 1—figure supplement 2E*). Further, we confirmed that the up-regulation of *Hox* genes was restricted to the pons in the *Rnf220*[+/-] mice (*Figure 1—figure supplement 2F*). In addition, RNA sequencing (RNA-seq) analysis of the pons also revealed an overall increase in *Hox* gene expression in adult *Rnf220*[+/-] mice (*Figure 1A*).

To identify the specific cell populations in which *Hox* genes were up-regulated, we conducted single-nucleus RNA-seq (snRNA-seq) analysis of the pons of adult wild-type (WT) and *Rnf220*[+/-] mice. In total, 125,956 and 122,266 cell transcriptomes with an average of approximately ≥700 genes for the WT and *Rnf220*[+/-] mice respectively were further analyzed. 15 cell clusters were identified by uniform manifold approximation and projection (UMAP) in both the WT and *Rnf220*[+/-] groups (*Figure 1B and C*). When we annotate these cell clusters by their uniquely and highly expressed markers (*Supplementary file 3*), we found that most of the clusters were identified as distinct neuronal groups (cluster_1, 2, 4, 6, 7, 8, 10, 11, 12, and 13), in addition to astrocyte (cluster_5), oligodendrocyte (cluster_3), oligodendrocyte precursor (cluster_14), ependymal cell (cluster_9), and a group not corresponding to any known cell types (cluster_0) (*Figure 1B*). Then, the *Hox* gene expression levels for each cell clusters were analyzed and found that up-regulation of *Hox* genes was most pronounced in three of these clusters (cluster_2, 7, and 10) (*Figure 1D*), albeit with distinct profiles. *Hox7-10* genes were up-regulated genes in cluster_2, while *Hox3-5* and *Hox1-3* genes were activated in cluster_7 and 10, respectively (*Figure 1D*). These findings indicate that *Hox* genes were specifically up-regulated in the pons of *Rnf220*[+/-] mice from late developmental stages on.

P19 embryonal carcinoma cells can be induced to differentiate and express *Hox* genes upon retinoic acid (RA) administration and have been used to study *Hox* regulation (*Vanderheyden and Defize, 2003Vanderheyden and Defize, 2003*; *Kondo et al., 1992*; *Pöpperl and Featherstone, 1993Pöpperl and Featherstone, 1993*; *Figure 1—figure supplement 3A*). Here, we tested the effects of *Rnf220* knockdown on RA-induced *Hox* gene activation in P19 cells. *Rnf220* knockdown further enhanced the activation of many *Hox* genes, including *Hoxa1*, *Hoxb1*, *Hoxa9*, and *Hoxb9*, by RA presence (*Figure 1E*), suggesting a general role of *Rnf220* in *Hox* regulation. Note that *Rnf220* knockdown had no clear effect on *Hox* expression in the absence of RA (*Figure 1—figure supplement 3B*).

### *Rnf220* insufficiency disturbs *Hox* expression pattern and the neuronal circuit formation in the PN

When we refer to the transcriptomic profiles of each cluster in the pons, we found that the cluster_10 likely represents cells derived from rhombomeres 2–5, given their expression of endogenous anterior *Hox* genes (*Hox1-3*), while the cluster_7 likely represents cells derived from rhombomeres 6–8, as evidenced by the expression of *Hox3-5* (*Figure 1D*). After mapping known neuron-specific markers to each cluster, PN-specific markers, including *Nfib*, *Pax6*, and *Barhl1*, were enriched in the cluster_7 (*Figure 2—figure supplement 1A*). The quantitative real-time polymerase chain reaction (qRT-PCR) results showed that compared to WT mice, some PN markers were up-regulated in *Rnf220*[+/-] mice

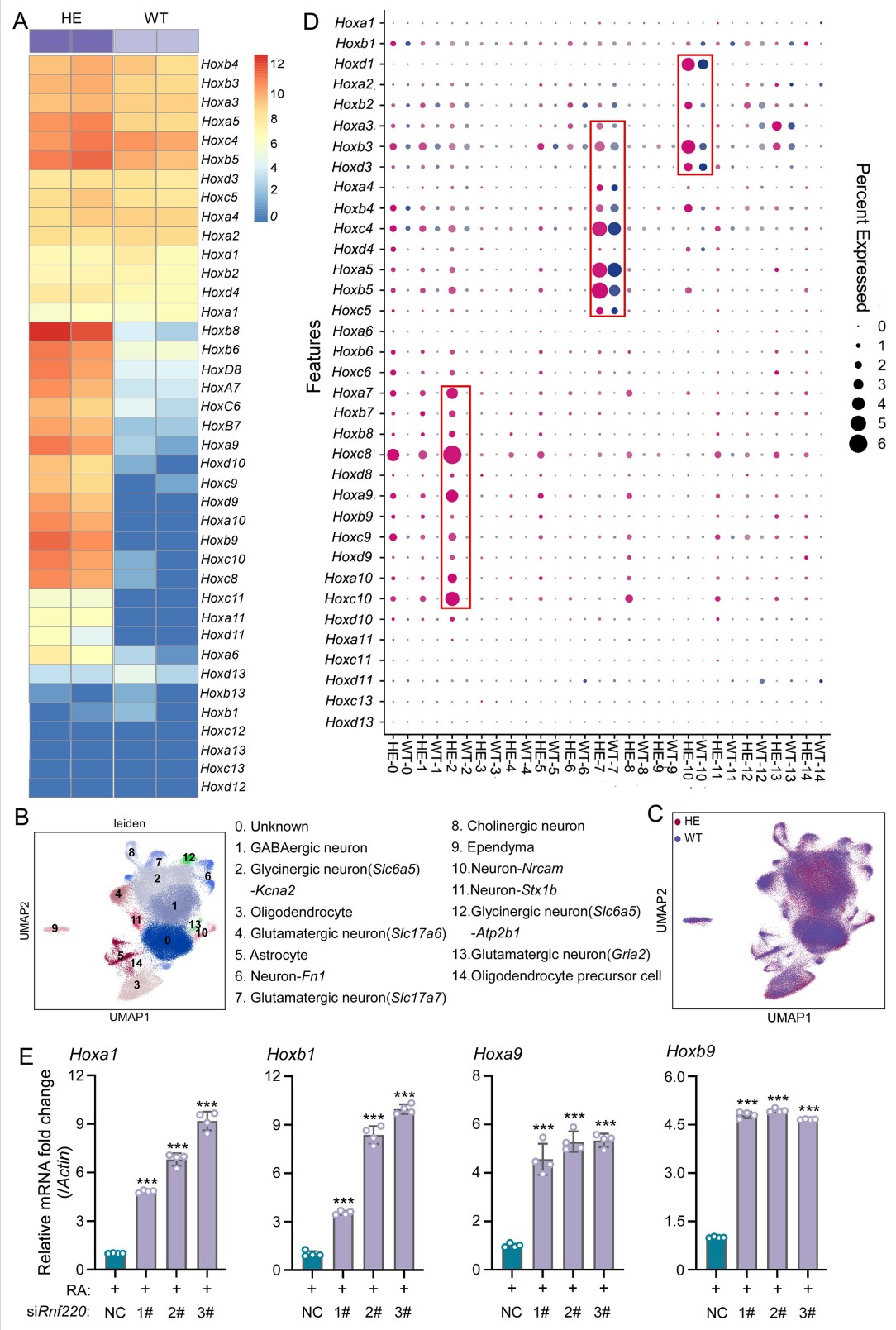

**Figure 1.** *Hox* genes up-regulated in pons of *Rnf220⁺ᐟ⁻* mice. (**A**) The heatmap of RNA sequencing (RNA-seq) data showing *Hox* genes expression in pons of WT or *Rnf220⁺ᐟ⁻* mice (n=2 mice per group). (**B–C**) Uniform manifold approximation and projection (UMAP) diagram showing 15 identified cell clusters annotated by single-nucleus RNA-seq (snRNA-seq) analysis of pons. Each dot represents a single cell, and cells are laid out to show similarities (n=3 mice per group). Genes in parentheses represent the marker genes of the cell group, while the genes following '-' represent the specific genes

*Figure 1 continued on next page*

*Figure 1 continued*

of this cell group. (**D**) Heatmap of snRNA-seq data showing *Hox* expression changes in each cell cluster. (**E**) Quantitative real-time polymerase chain reaction (*qRT-PCR*) analysis showing mRNA levels of indicated *Hox* genes in P19 cells when endogenous *Rnf220* was knocked down by small interfering RNAs (siRNAs) in the presence of RA. WT, wild-type; HE, heterozygote; RA, retinoic acid. **p<0.01, ***p<0.001.

The online version of this article includes the following figure supplement(s) for figure 1:

**Figure supplement 1.** *Hox* genes exhibited de-repression in hindbrain of *Rnf220⁻/⁻* embryos at late developmental stages.

**Figure supplement 2.** *Hox* genes were up-regulated in brainstem of *Rnf220⁺/⁻* mice.

**Figure supplement 3.** Expression levels of *Hox* genes were not affected by *Rnf220* knockdown in P19 cell line without RA induction.

(*Figure 2—figure supplement 1B*). Originating at the posterior rhombic lip and migrating tangentially to their final position in the ventral part of the pons, PN neurons serve as relay cells between the cerebral cortex and cerebellum (*Di Meglio et al., 2013*; *Kratochwil et al., 2017*; *Maheshwari et al., 2020*). The expression pattern of *Hox* genes is crucial for the migration and the final neuronal circuit formation of the PN neurons (*Kratochwil et al., 2017*; *Maheshwari et al., 2020*). We analyzed the expression pattern of the endogenous *Hox* genes, including *Hox3-5*, in the PN neurons and the connection between the cortex and the PN neurons in *Rnf220⁺/⁻* mice (*Figure 2*; *Figure 2—figure supplement 1C*). The results of RT-PCR assays indicated an up-regulation in *Hox4-5* expression levels in the *Rnf220⁺/⁻* PN (*Figure 2—figure supplement 1C*). In the PN, endogenous *Hox3-5* genes exhibit a nested and unique expression pattern along the rostral-caudal axis, with *Hox3* ubiquitously expressed throughout the PN, *Hox4* localized to the middle and posterior regions, and *Hox5* confined to the posterior segment (*Kratochwil et al., 2017*). Next, the PN was evenly dissected into the rostral, middle, and caudal segments along the rostral-caudal axis and the endogenous expression pattern of *Hox* genes was examined in each section. The results showed that all the *Hox3-5* paralogs were uniformly up-regulated along the rostral-caudal axis in the *Rnf220⁺/⁻* PN (*Figure 2A*).

There is no difference of structure in PN or even pons between WT and *Rnf220⁺/⁻* mice (*Figure 2—figure supplement 2A and B*; *Figure 2C–E*). Considering the motor deficits of *Rnf220⁺/⁻* mice (*Ma et al., 2021*; *Ma et al., 2019*), the neuronal circuit between the motor cortex and the PN neurons were traced anterogradely by a non-transneuronal tracing virus (rAAV-hSyn-EGFP-WPRE-hGH-polyA) (*Figure 2B*). The results showed that fewer PN neurons were targeted by axons from the motor cortex in the *Rnf220⁺/⁻* mice (*Figure 2C and F–G*; *Figure 2—figure supplement 2C*). In addition, the projection from the motor cortex was centralized in the PN of *Rnf220⁺/⁻* mice (*Figure 2C*; *Figure 2—figure supplement 2C*).

Taken together, both the expression level/pattern of the endogenous *Hox* genes and the neuronal projection pattern from the motor cortex were disturbed by *Rnf220* insufficiency in the mouse PN.

## RNF220 targets WDR5 for K48-linked polyubiquitination and degradation

The up-regulation of *Hox* genes in the *Rnf220⁺/⁻* mice suggested a role for RNF220 in the maintenance of *Hox* expression pattern. Generally, the *Hox* expression pattern is maintained epigenetically by the PcG and trxG complexes, which regulate the silencing and activation of *Hox* genes, respectively (*Bahrampour et al., 2019*; *Chopra et al., 2009*; *Kang et al., 2022*; *Papp and Müller, 2006*). Indeed, we did observe the local epigenetic modification changes, i.e., the down-regulated H3K27me3 signals and up-regulated H3K4me3 signals, in some *Hox* cluster in hindbrain of *Rnf220⁺/⁻* mice by ChIP-qPCR assays (*Figure 3—figure supplement 1A and B*). Therefore, protein levels of the core components of PcG and TrxG complexes in the mouse hindbrain and pons were examined and the results showed that although the protein levels of the core components of PcG we examined were comparable between WT and *Rnf220⁺/⁻*, a clear increase in the protein level of WDR5, a key component of the trxG complex, was observed in the hindbrain of both *Rnf220⁺/⁻* and *Rnf220⁻/⁻* mouse embryos at E18.5 (*Figure 3—figure supplement 2A and B*; *Figure 3A*), the phenomenon was also observed at E16.5 (*Figure 3B*). We next tested the expression of WDR5 in the mouse pons, cortex, and cerebellum at adult and found that the increase in the protein level of WDR5 was only observed in the pons, but not in the cortex or cerebellum, in *Rnf220⁺/⁻* mice (*Figure 3C and E*; *Figure 3—figure supplement 2C and D*), which is consistent with the pons-specific up-regulation in *Hox* genes expression. In addition, the protein level

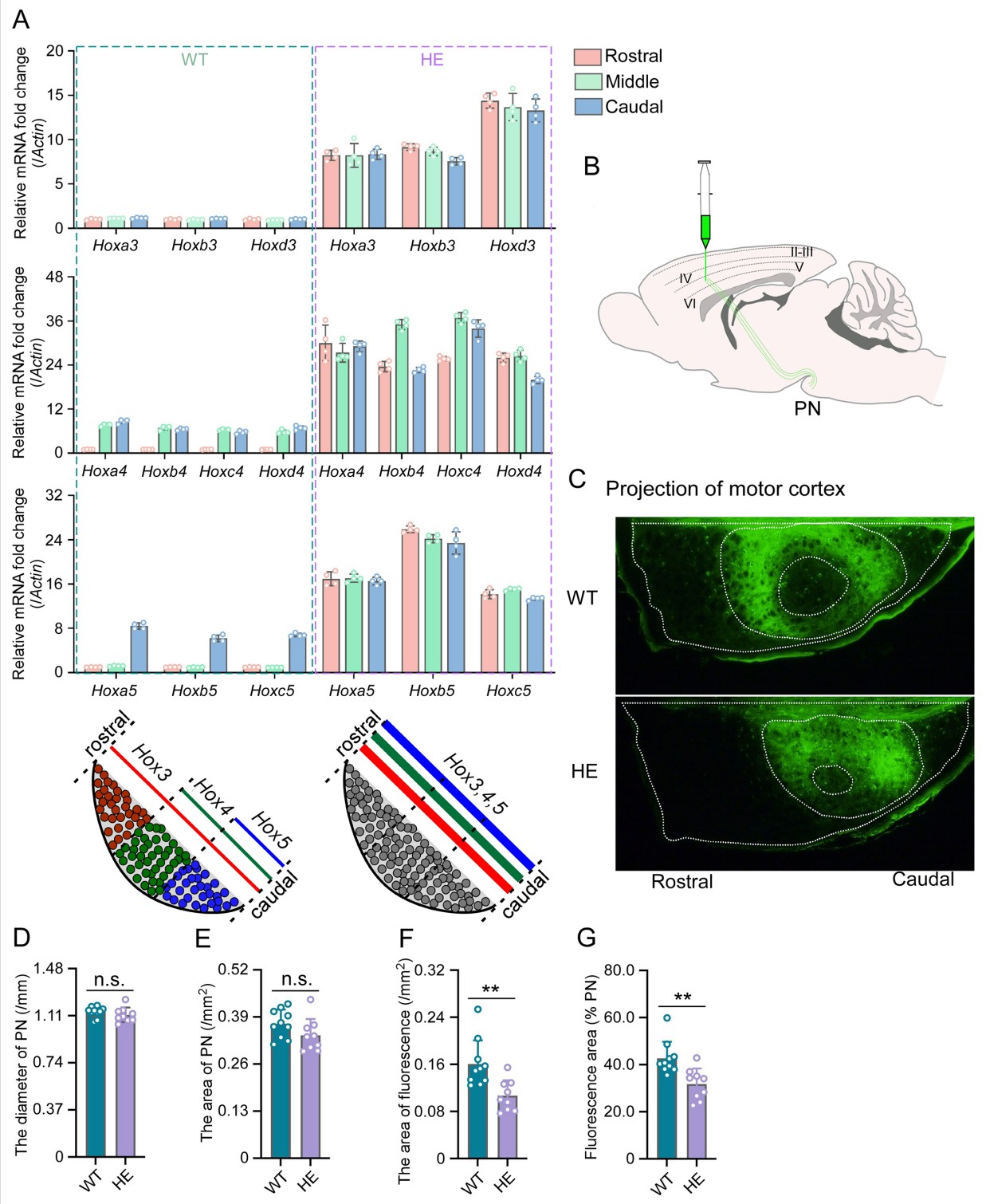

**Figure 2.** *Hox* gene expression was dysregulated and motor cortex projections were disorganized in pontine nuclei (PN) of *Rnf220*+/- mice.
(**A**) Quantitative real-time polymerase chain reaction (qRT-PCR) analysis of relative expression levels of *Hox3*, *Hox4*, and *Hox5* in rostral, middle, and caudal sections of PN in WT and *Rnf220*+/- mice. Expression level of each gene in rostral section of WT PN was set to 1 (n=5 mice per group). Bar graphs show the relative levels normalized against rostral group in the respective wild-type mice. (**B**) Diagram of experimental stereotactic injections. (**C**) Green fluoresce showed the projection from motor cortex to PNs in adult (2 months) WT and *Rnf220*+/- mice (n=10 in WT group and n=9 in *Rnf220*+/- group). (**D–E**) The diameter (**D**) and area (**E**) sizes of PN in WT and *Rnf220*+/- mice. Each data presents the average diameter and area sizes of PN from four consecutive slices for completely presenting circular fluorescence projections. (**F**) The area of fluorescence projection from motor cortex to PN in WT

*Figure 2 continued on next page*

*Figure 2 continued*

and *Rnf220⁺ᐟ⁻* mice. The sample used for statistics is consistent with the one selected in D–E. Each data represents the average fluorescence area of four consecutive slices. (**G**) The proportion of projected fluorescence area from motor cortex to PN area. The sample used for statistics is consistent with the one selected in D–E. WT, wild-type; HE, heterozygote. n.s., not significant. **p<0.01.

The online version of this article includes the following figure supplement(s) for figure 2:

**Figure supplement 1.** *Hox* genes were up-regulated in pontine nuclei (PN) of *Rnf220⁺ᐟ⁻* mice.

**Figure supplement 2.** PN showed no structure difference but disorganized projection pattern from motor cortex between WT and *Rnf220⁺ᐟ⁻* mice.

of WDR5 was also enhanced in the PN of *Rnf220⁺ᐟ⁻* mice (**Figure 3D**) and *Rnf220* knockdown P19 cells in the presence of RA (**Figure 3F**).

The above data suggest that WDR5 might be a direct target of RNF220 for polyubiquitination and thus degradation. Indeed, in HEK293 cells transiently co-transfected with FLAG-tagged RNF220 and myc-tagged WDR5, WDR5 co-immunoprecipitated with RNF220 (**Figure 4A**). In the reverse experiment, RNF220 also co-immunoprecipitated with WDR5 (**Figure 4B**). Furthermore, in the brainstem,

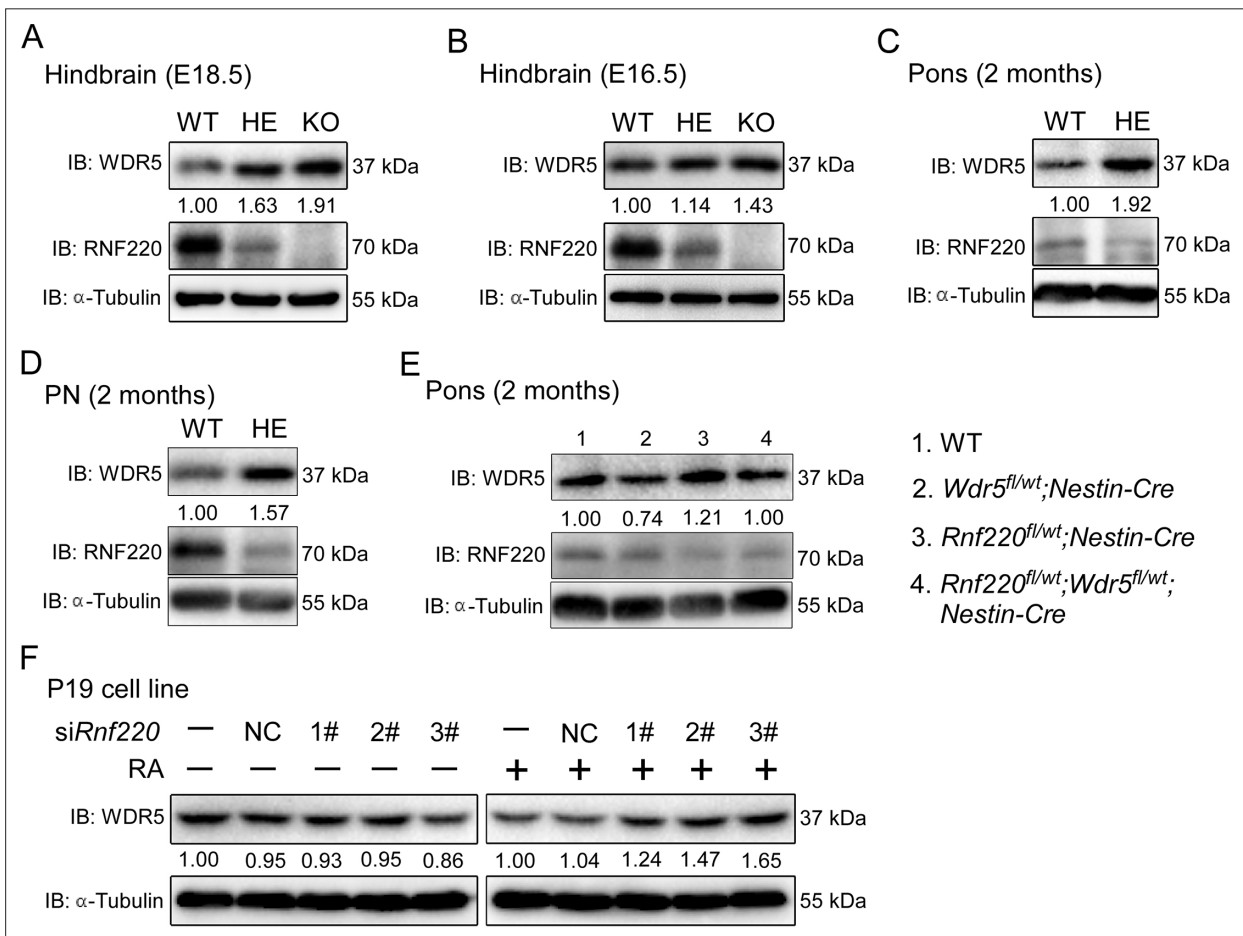

**Figure 3.** RNF220 mediates WDR5 degradation. (**A–D**) Western blots analysis showing the protein level of WDR5 in the indicated brain tissues of mice with different genotypes at different ages. (**E**) Western blot analysis showing WDR5 levels in the pons of adult mice with indicated genotypes. (**F**) Western blot analysis of protein levels of WDR5 in P19 cells with *Rnf220* knockdown or not in the presence or absence of RA. IB, immunoblot; WT, wild-type; HE, heterozygote; KO, knockout; PN, pontine nuclei; NC, negative control; RA, retinoic acid.

The online version of this article includes the following figure supplement(s) for figure 3:

**Figure supplement 1.** Repressive epigenetic modification was down-regulated while activated epigenetic modification was up-regulated in promoter regions of indicated *Hox* genes in hindbrains of *Rnf220⁺ᐟ⁻* mice.

**Figure supplement 2.** Protein levels of the indicated core components of PRC1 and PRC2 complex in indicated mouse brain tissues of different genotypes.

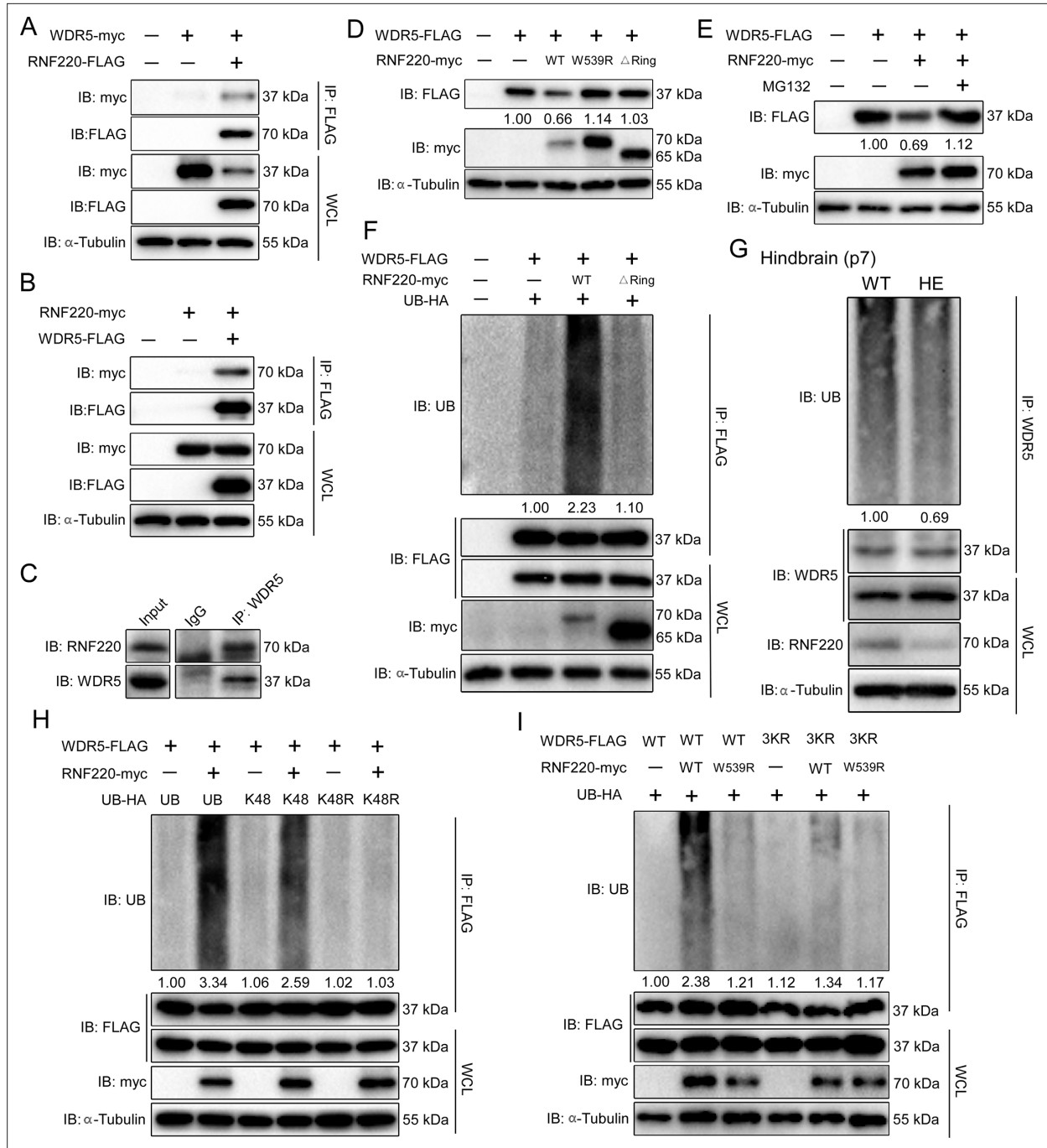

**Figure 4.** RNF220 interacts with and targets WDR5 for K48-linked polyubiquitination. (**A–B**) Co-immunoprecipitation (co-IP) analysis of interactions between RNF220 and WDR5 in HEK293 cells. HEK293 cells were transfected with indicated plasmids and harvested after 48 hr. Cell lysates were immunoprecipitated with anti-FLAG beads. Whole-cell lysate and immunoprecipitates were subjected to western blot analysis using indicated antibodies. (**C**) Endogenous co-immunoprecipitation analysis showing the interaction between RNF220 and WDR5 in hindbrains of WT mice. (**D**) Western blots analysis shows the protein level of WDR5 when co-expressed with wild-type or mutated RNF220 in HEK293 cells. (**E**) Western blots analysis shows the protein level of WDR5 when co-expressed with RNF220 in HEK293 cells in the presence of MG132 (10 mM) or not. (**F**) In vivo ubiquitination assays showing the ubiquitination status of WDR5 when co-expressed with WT or mutated RNF220 in HEK293 cells. (**G**) In vivo ubiquitination assays showing the ubiquitination status of WDR5 in hindbrains of WT and *Rnf220*^+/- mice. (**H**) In vivo ubiquitination assays showing RNF220-induced polyubiquitination of WDR5 when the indicated ubiquitin mutations were used in HEK293 cells. (**I**) In vivo ubiquitination assays showing the ubiquitination status of the indicated WDR5 mutants when co-expressed with WT or ligase-dead RNF220 in HEK293 cells. WT, wild-type; HE, heterozygote; KO, knockout; IB, immunoblot; IP, immunoprecipitation; UB, ubiquitin; WCL, whole-cell lysate; △Ring, RNF220 Ring domain deletion;

*Figure 4 continued on next page*

*Figure 4 continued*

W539R, RNF220 ligase dead mutation; K48, ubiquitin with all lysines except the K48 mutated to arginine; K48R, ubiquitin with the K48 was substituted by an arginine; 3KR, substitution of lysines at the positions of 109, 112, and 120 in WDR5 with arginines simultaneously.

The online version of this article includes the following figure supplement(s) for figure 4:

**Figure supplement 1.** RNF220 interacted with and targeted WDR5 for polyubiquitination at multiple lysine sites.

when WDR5 was immunoprecipitated using an anti-WDR5 antibody, endogenous RNF220 was also detected in the immunoprecipitate. Then we examined if the WDR5 protein level was regulated by RNF220. The results showed that co-expression of WT RNF220, but not the ligase-dead mutant (W539A) or the RING domain deletion (△Ring) form, clearly reduced the protein level of WDR5 in HEK293 cells (*Figure 4D*). In addition, the reduction in the protein level of WDR5 by RNF220 overexpression was blocked by MG132, suggesting a role for the proteasome in this regulation (*Figure 4E*).

We further carried out polyubiquitination assays to examine if WDR5 may be a target for RNF220. It was found that co-expression of RNF220 strongly enhanced the polyubiquitination of WDR5 protein, and this regulation depends on the E3 ubiquitin ligase activity of RNF220 because RNF220△Ring failed to promote the polyubiquitination of WDR5 (*Figure 4F*). Moreover, when the endogenous WDR5 protein was immunoprecipitated from the mouse pons, its polyubiquitination level in *Rnf220*$^{+/-}$ mice was markedly decreased compared to WT mice (*Figure 4G*).

Different types of ubiquitination linkages have distinct regulatory effects on the stability or activity of protein targets, and K48-linked polyubiquitination usually leads to proteasomal degradation of protein targets (*Grice and Nathan, 2016Grice and Nathan, 2016*.). Indeed, WDR5 were only ubiquitinated by RNF220 when K48-type ubiquitin was present (*Figure 4H*). Consistent with this, when the K48R mutated ubiquitin was co-expressed, the RNF220-induced polyubiquitination level of WDR5 protein was fully diminished (*Figure 4H*). To determine the exact lysines ubiquitinated by RNF220, we first tested the effects of RNF220 on the stability of different WDR5 truncates and found that the truncate remaining 1-127aa was enough to be degraded by RNF220, suggesting the corresponding lysines were included in this region (*Figure 4—figure supplement 1A*). We individually mutated all these conserved lysines into arginines and then examined RNF220-mediated ubiquitination of each mutant. It was found that K109, K112, and K120 were required for the polyubiquitination of WDR5 (*Figure 4—figure supplement 1B*). Furthermore, when these lysine residues were simultaneously mutated into arginines, RNF220 failed to enhance the polyubiquitination levels of the resulted WDR5-3KR mutants (*Figure 4I*), suggesting that these lysine residues are direct ubiquitination sites. Together, these results indicate that RNF220 regulates WDR5 ubiquitination by adding K48-linked polyubiquitin chains at the lysine sites of K109, K112, and K120.

## The maintenance of *Hox* expression by RNF220 depends on its regulation to WDR5

To investigate the requirement of the RNF220-mediated regulation to WDR5 in *Hox* expression maintenance by RNF220, we first examined whether *Wdr5* knockdown could mitigate the impact of *Rnf220* knockdown on *Hox* expression in P19 cells in the presence of RA. It is observed that the stimulation of *Hox* genes by *Rnf220* knockdown was significantly reduced when *Wdr5* was knocked down simultaneously by small interfering RNAs (siRNAs) in the presence of RA (*Figure 5A–C*). Note that the expression levels of *Hox* genes showed no clear changes when without RA or only *Wdr5 was* knocked down in P19 cells (*Figure 5—figure supplement 1A and B*).

Using in utero microinjection, we tested the effects of WDR5 inhibition by WDR5-IN-4, an established WDR5 inhibitor (*Aho et al., 2019*), on the up-regulation of *Hox* genes in hindbrains of the *Rnf220*$^{+/-}$ mice at late embryonic stages (*Figure 6A*). It was found that the up-regulation of *Hox3-5* genes in the hindbrain of the *Rnf220*$^{+/-}$ embryos was largely reversed by WDR5-IN-4 (*Figure 6C*). Notably, we found that WDR5-IN-4 injection had no effect on the endogenous *Rnf220* expression in the hindbrain (*Figure 6B*). Last, the effect of genetic ablation of *Wdr5* in neural system on the up-regulation of *Hox* genes in RNF220 hypoinsufficient mice was examined. Considering *Wdr5*$^{+/-}$ mice is embryonic lethal, and the up-regulation of *Hox* genes could also be observed in the hindbrains of *Rnf220*$^{fl/wt}$;*Nestin-Cre* mice, we used *Rnf220*$^{fl/wt}$;*Nestin-Cre* and *Wdr5*$^{fl/wt}$ mice to generate WT, *Rnf220*$^{fl/wt}$;*Nestin-Cre*, *Wdr5*$^{fl/wt}$;*Nestin-Cre*, and *Rnf220*$^{fl/wt}$;*Wdr5*$^{fl/wt}$;*Nestin-Cre* mice (*Figure 6D*). It was found

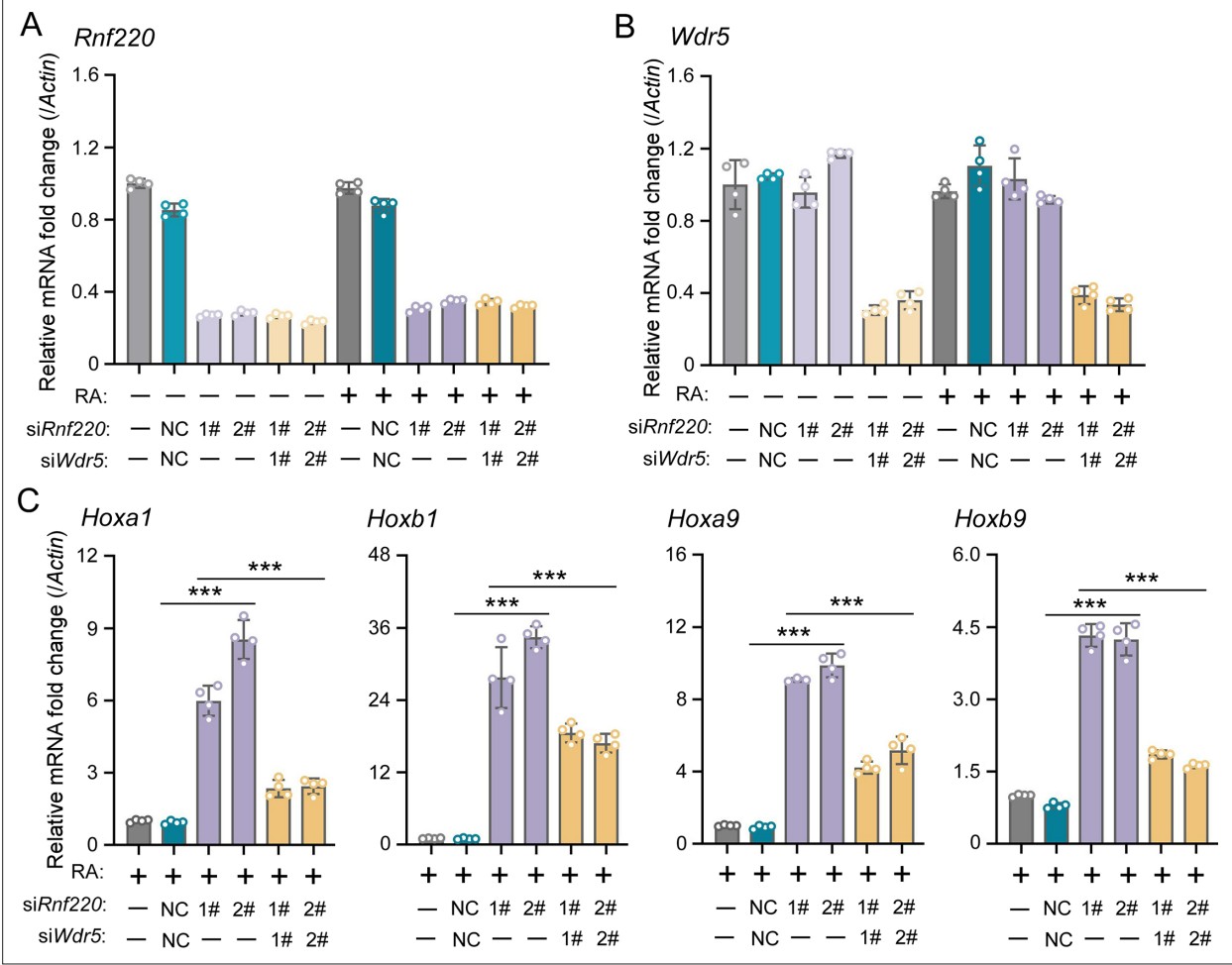

**Figure 5.** WDR5 recovered *Rnf220* deficiency-induced up-regulation of *Hox* genes in P19 cell line. (**A–B**) Quantitative real-time polymerase chain reaction (qRT-PCR) analysis showing the expression levels of *Rnf220* (**A**) and *Wdr5* (**B**) when transfected the indicated combinations of small interfering RNAs (siRNAs) against *Rnf220* or *Wdr5* in the presence or absence of RA. Bar graphs show the relative levels normalized against control group without siRNA or RA treatment. (**C**) qRT-PCR analysis showing the expression levels of Hoxa1, Hoxb1, Hoxa9, Hoxb9 when siRnf220, together with siWDR5 or not, were transfected in P19 cells treated with RA. RA, retinoic acid. n.s., not significant; ***p<0.001.

The online version of this article includes the following figure supplement(s) for figure 5:

**Figure supplement 1.** *Wdr5* knockdown had no effect on *Hox* genes expression in P19 cells in the presence of RA or not.

that the up-regulation of *Hox3-5* genes in the pons of *Rnf220*^fl/wt^;*Nestin-Cre* mice were markedly recovered by *Wdr5* genetic ablation in *Rnf220*^fl/wt^;*Wdr5*^fl/wt^;*Nestin-Cre* mice (*Figure 6E*). ChIP-qPCR also showed the local epigenetic modification changes in some *Hox* cluster, with H3K27me3 up-regulated and H3K4me3 down-regulated in *Rnf220* and *Wdr5* double knockdown P19 cell line related to *Rnf220* single knockdown P19 cell line in the presence of RA (*Figure 6—figure supplement 1*).

Taken together, *Rnf220* dificiency induces the up-regulation of *Hox* genes in the mouse hindbrain and thus a defects in cortex-PN nueronal circuits. Mechanistically, we found that RNF220 regulates the protein stability of WDR5 via ubiquitination in the mouse hindbrain. WDR5 conditional knockdown or functional inhibition reversed the up-regulation of *Hox* genes expression in *Rnf220* deficient mice. The above findings support the involvement of WDR5 regulation by RNF220 in the maintenance of *Hox* expression pattern in the mouse hindbrain.

## Discussion

During early embryonic development, the *Hox* genes are collinearly expressed in the hindbrain and spinal cord along the A-P axis to guide regional neuronal identity. Later on, the segmental *Hox* gene

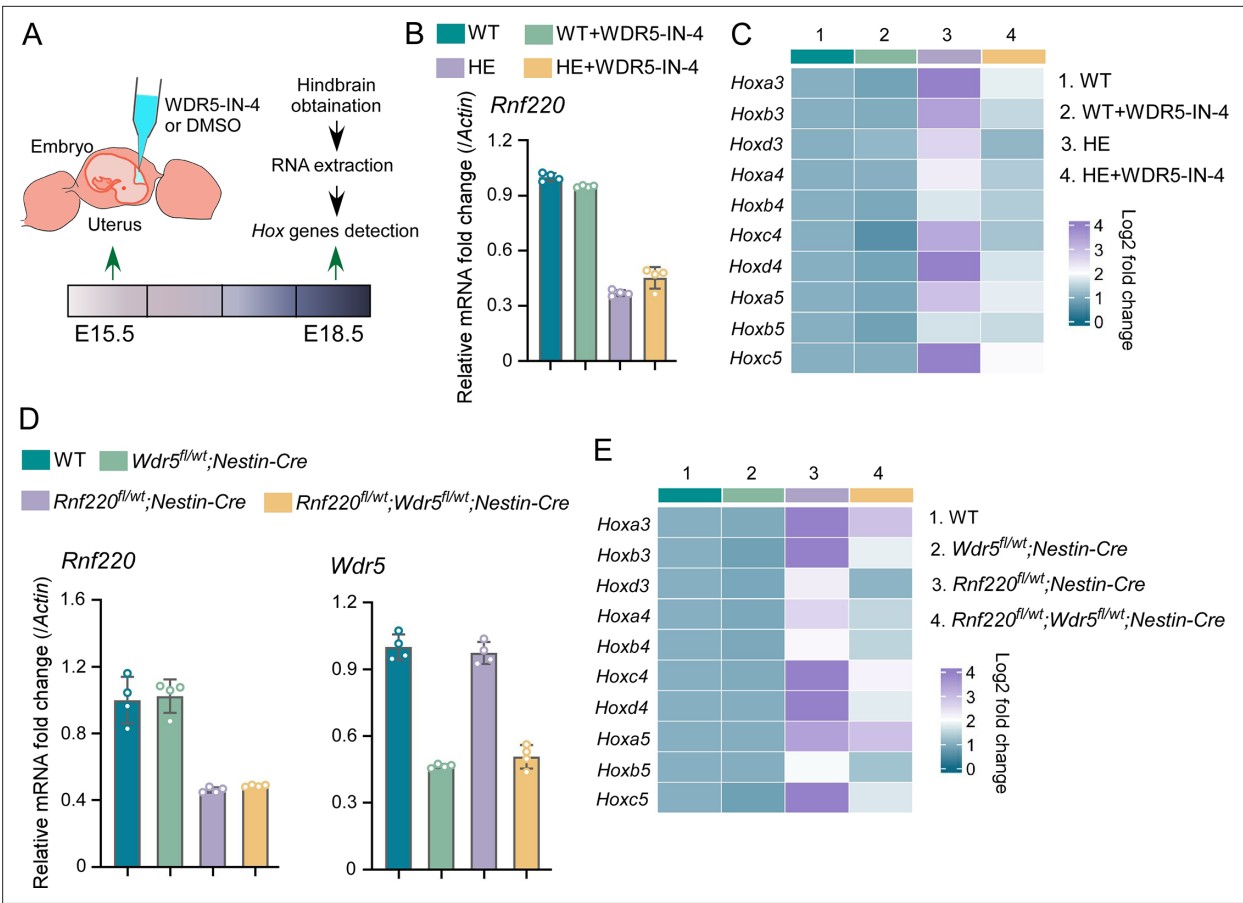

**Figure 6.** Genetic and pharmacological ablation of WDR5 recovered *Rnf220* deficiency-induced up-regulation of *Hox* genes. (**A**) Diagram of experimental strategy for in utero local injection of WDR5 inhibitors. (**B–C**) Quantitative real-time polymerase chain reaction (qRT-PCR) analysis of expression levels of *Rnf220* (**B**), *Hox3-Hox5* (**C**) in hindbrains of WT and *Rnf220*^(+/-) mouse embryos treated with WDR5 inhibitors or not at E18.5 (n=3 mice per group). *Actin* was used as the internal controls (**C**). Heatmap of *Hox* expression showed the relative levels normalized against WT group. (**D–E**) qRT-PCR analysis of expression levels of *Rnf220* (**D**), *Wdr5* (**D**), *Hox3-Hox5* (**E**) in pons of P15 mice with indicated genotypes (n=2 mice per group). *Gapdh* was used as the internal controls (**E**). Heatmap of *Hox* expression showed the relative levels normalized against WT group. WT, wild-type; HE, heterozygote.

The online version of this article includes the following figure supplement(s) for figure 6:

**Figure supplement 1.** *Rnf220* and *Wdr5* co-suppression recovered *Hox* epigenetic modification to a certain degree.

expression pattern in the hindbrain is maintained till at least early postnatal stages. The *Hox* genes are also expressed in adult hindbrains with restricted anterior boundaries (*Philippidou and Dasen, 2013*; *Miller and Dasen, 2024*; *Di Bonito et al., 2013*). In addition to their roles in progenitor cell specification, cell survival, neuronal migration, axon guidance, and dendrite morphogenesis during early development stages (*Smith and Kratsios, 2024*), *Hox* genes also play key roles in the regulation of synapse formation, neuronal terminal identity, and neural circuit assembly at late stages (*Feng et al., 2020*; *Feng et al., 2021*; *Philippidou and Dasen, 2013*). Our study revealed a dose-dependent role of RNF220 and WDR5 in the maintenance of *Hox* expression in the hindbrain, which might have a functional role in the neural circuit organization of the pons in mice.

In mammals, WDR5, a key component of the COMPASS-related complexes, has been reported to be absolutely required for the regulation of *Hox* genes expression during mammalian embryo development and cancer progression via different mechanisms (*Wysocka et al., 2005*; *Chen et al., 2021*; *Yu et al., 2021*). Here, we report a WDR5 protein stability controlling mechanism by RNF220-mediated polyubiquitination and illustrate the role of this regulation on the maintenance of *Hox* genes expression in the mouse hindbrain. The stabilization of WDR5 and stimulation of *Hox* expression upon RNF220 knockdown was also recapitulated in RA-treated P19 cells, a cellular model for *Hox* regulation study.

**Table 1.** Primers used for quantitative real-time polymerase chain reaction (qRT-PCR).

| Genes | Forward primers | Reverse primers |
|---|---|---|
| Actin | GCCAACCGTGAAAAGATGAC | GAGGCATACAGGGACAGCAC |
| Gapdh | AGGTCGGTGTGAACGGATTTG | TGTAGACCATGTAGTTGAGGTCA |
| Rnf220 | GTCTCAGTAGACAAGGACGTTCACA | GGGGTGGAGGTGTAGTAAGGAAG |
| Wdr5 | CGTGACAGGCGGGAAGTGGA | CGGGTGACAAGCCGTGGAAAT |
| Hoxa1 | AGCTCTGTGAGCTGCTTGGT | AAAAGAAACCCTCCCAAAACA |
| Hoxa2 | TGCCATCAGCTATTTCCAGG | GATGAAGGAGAAGAAGGCGG |
| Hoxa3 | TCTTAACATGGAGGGAGCCA | TCTGAAGGCTACGTGTGCTG |
| Hoxa4 | ACGCTGTGCCCCAGTATAAG | ACCTTGATGGTAGGTGTGGC |
| Hoxa5 | CAGGGTCTGGTAGCGAGTGT | CTCAGCCCCAGATCTACCC |
| Hoxa6 | GTCTGGTAGCGCGTGTAGGT | CCCTGTTTACCCCTGGATG |
| Hoxa7 | CTTCTCCAGTTCCAGCGTCT | AAGCCAGTTTCCGCATCTAC |
| Hoxa9 | GTAAGGGCATCGCTTCTTCC | ACAATGCCGAGAATGAGAGC |
| Hoxa10 | TCTTTGCTGTGAGCCAGTTG | CTCCAGCCCCTTCAGAAAAC |
| Hoxa11 | CCTTTTCCAAGTCGCAATGT | AGGCTCCAGCCTACTGGAAT |
| Hoxa13 | CGGTGTCCATGTACTTGTCG | AGCGGCTACTACCCGTGC |
| Hoxb1 | GGTGAAGTTTGTGCGGAGAC | TTCGACTGGATGAAGGTCAA |
| Hoxb2 | GAACCAGACTTTGACCTGCC | GAGCTGGAGAAGGAGTTCCA |
| Hoxb3 | ATCTGTTTGGTGAGGGTGGA | CCGCACCTACCAGTACCACT |
| Hoxb4 | GACCTGCTGGCGAGTGTAG | CTGGATGCGCAAAGTTCAC |
| Hoxb5 | CTGGTAGCGAGTATAGGCGG | AGGGGCAGACTCCACAGATA |
| Hoxb6 | TAGCGTGTGTAGGTCTGGCG | AGCAGAAGTGCTCCACGC |
| Hoxb7 | CTTTCTCCAGCTCCAGGGTC | AACTTCCGGATCTACCCCTG |
| Hoxb8 | GAACTCCTTCTCCAGCTCCA | CACAGCTCTTTCCCTGGATG |
| Hoxb9 | TCCAGCGTCTGGTATTTGGT | GAAGCGAGGACAAAGAGAGG |
| Hoxb13 | TGCCCCTTGCTATAGGGAAT | ATTCTGGAAAGCAGCGTTTG |
| Hoxc4 | CTAATTCCAGGACCTGCTGC | AAAAATTCACGTTAGCACGGT |
| Hoxc5 | TTCTCGAGTTCCAGGGTCTG | ATTTACCCGTGGATGACCAA |
| Hoxc6 | CAGGGTCTGGTACCGAGAGTA | TCCAGATTTACCCCTGGATG |
| Hoxc8 | CAAGGTCTGATACCGGCTGT | ATCAGAACTCGTCTCCCAGC |
| Hoxc9 | AATCTGTCTCTGTCGGCTCC | AGTCTGGGCTCCAAAGTCAC |
| Hoxc10 | ACCTCTTCTTCCTTCCGCTC | ACTCCAGTCCAGACACCTCG |
| Hoxc11 | AAATGAAGGCTCCTACGGCG | TGTCGAAGAAGCGGTCGAAA |
| Hoxc12 | AATACGGCTTGCGCTTCTT | GACCCTGGCTCTCTGGTTTC |
| Hoxc13 | CTCACTTCGGGCTGTAGAGG | TCAGGTGTACTGCTCCAAGG |
| Hoxd1 | TCTGTCAGTTGCTTGGTGCT | TGAAAGTGAAGAGGAACGCC |
| Hoxd3 | ACCAGCTGAGCACTCGTGTA | AGAACAGCTGTGCCACTTCA |
| Hoxd4 | CTCCCTGGGCTGAGACTGT | CCCTGGGAACCACTGTTCT |
| Hoxd8 | GCCCGCGAAGTTTTACGGAT | TAAGTGGTCTGGGTCCTCGC |
| Hoxd9 | TTGTTTGGGTCAAGTTGCTG | CTCAGCTTGCAGCGATCA |
| Hoxd10 | TCTCCTGCACTTCGGGAC | GGAGCCCACTAAAGTCTCCC |
| Hoxd11 | AGTGAGGTTGAGCATCCGAG | ACACCAAGTACCAGATCCGC |
| Hoxd12 | TGCTTTGTGTAGGGTTTCCTCT | CTTCACTGCCCGACGGTA |

*Table 1 continued on next page*

*Table 1 continued*

| Genes | Forward primers | Reverse primers |
|-------|-----------------|-----------------|
| *Hoxd13* | TGGTGTAAGGCACCCTTTTC | CCCATTTTTGGAAATCATCC |
| *Unc5b* | CGTGACAGGCGGGAAGTGGA | CGGGTGACAAGCCGTGGAAAT |
| *Pax6* | AGACAACAACAAAGCGGACT | CTTCGCAAATGACAACTGAC |
| *Barhl1* | TACCAGAACCGCAGGACTAAAT | AGAAATAAGGCGACGGGAACAT |
| *Nfib* | AGAAGCCCGAAATCAAGCAG | CCAGTCACGGTAAGCACAAA |
| *Neph2* | ACAAGGTTCGGAAATGAAGTCG | GTTGCCATTAGGACGAGGAA |

Interestingly, when examined at single cell level, different *Hox* stimulation profiles were detected in different neuronal groups, so that *Hox1-3* were up-regulated in cluster 10, while *Hox3-5* and *Hox7-10* up-regulation appeared in cluster 7 and cluster 2, respectively (*Figure 1D*). Clusters 10 and 7 express endogenous *Hox1-3* and *Hox3-5*, respectively. It is reasonable that the expression of these *Hox* genes is stimulated upon WDR5 elevation due to RNF220 insufficiency. Cluster 2 is most interesting in that, *Hox7-10* were up-regulated in these cells which were not expressed in WT hindbrain. This cluster was defined as glycinergic neuron cells by specific *Slc6a5* expression. The nature of this group of cells and the phenotypic effect of *Hox* up-regulation await further analysis. The de-regulation of *Hox* genes might have wide effect on the development of many nuclei in the pons, including cell differentiation and neural circuit formation. For the PN, we showed here that the patterning and projection pattern from the motor cortex were affected in the *Rnf220⁺ᐟ⁻* mice.

The regulation of WDR5 and *Hox* expression by RNF220 seems to be context dependent and precisely controlled in vivo, depending on the molecular and epigenetic status of the cell. For example, although WDR5 and RNF220 are widely co-expressed in the brain, WDR5 levels are not affected in most regions in the *Rnf220⁺ᐟ⁻* brain, including the cortex, cerebellum, etc. (*Figure 3—figure supplement 2C and D*). This is also true for EED, another epigenetic regulator, which is targeted by RNF220 in the cerebellum, but not in the pons (*Ma et al., 2020*). In untreated P19 cells, knockdown of RNF220 also has no effect on the stability of endogenous WDR5, although overexpression of RNF220 did reduce WDR5 level in 293T cells. We suggest that RA treatment of P19 cells helps to set up the molecular environment that facilitates *Hox* expression as well as RNF220-WDR5 interaction.

In summary, our data support WDR5 as a RNF220 target involved in the maintenance of *Hox* expression and thus development of the pons.

# Materials and methods
## Mouse strains and genotyping

All procedures involving mice were conducted in accordance with the guidelines of the Animal Care and Use Committee of the Kunming Institute of Zoology, Chinese Academy of Sciences (IACUC-PA-2021-07-018). Mice were housed under standard conditions at a temperature range of 20–22°C, humidity of 30–45%, and 12 hr light/dark cycle. *Rnf220* floxed mice (*Rnf220ᶠˡᐟᶠˡ*) were originally 129Sz/SvPasCrl background and mated with C57BL/6 then. Other mice used were maintained on a C57BL/6 background. *Vasa-Cre* mice were used to mate with *Rnf220* floxed mice (*Rnf220ᶠˡᐟᶠˡ*) to generate *Rnf220* germ cell knockout (*Rnf220⁻ᐟ⁻*) or heterozygote (*Rnf220⁺ᐟ⁻*) mice. *Nestin-Cre* mice were used to mate with *Rnf220ᶠˡᐟᶠˡ* and *Wdr5* floxed mice (*Wdr5ᶠˡᐟᶠˡ*) to generate conditional neural specific knockout or heterozygote mice. *Rnf220ᶠˡᐟʷᵗ;Wdr5ᶠˡᐟʷᵗ;Nestin-Cre* mice were obtained by crossing *Rnf220ᶠˡᐟʷᵗ;Nestin-Cre* with *Wdr5ᶠˡᐟʷᵗ* mice.

Genotypes were determined by PCR using genome DNA from tail tips as templates. PCR primers were listed as follows: 5'-CTG TTG ATG AAG GTC CTG GTT-3' and 5'-CAG GAA AAT CAA TAG ACA ACT T-3' were used to detect *Rnf220* floxP carrying. 5'-CTG TTG ATG AAG GTC CTG GTT-3' and 5'-CTG ATT TCC AGC AAC CTA AA-3' were used to detect *Rnf220* knockout. 5'-GCC TGC ATT ACC GGT CGA TGC-3' and 5'-CAG GGT GTT ATA GCA ATC CC-3' were used for *Cre* positive detection. 5'-GAA TAA CTA CTT TCC CTC AGA CC-3' and 5'-CAG GCC AAG TAA CAG GAG GTA G-3' were used to detect *Wdr5* floxP carrying. 5'-GAA TAA CTA CTT TCC CTC AGA CC-3' and 5'-AGA CCC TGA GTG AGG ATA CAT AA-3' were used to detect *Wdr5* knockout.

## Cell culture

The HEK293 cell line was from Conservation Genetics CAS Kunming Cell Bank (KCB 200408YJ) and P19 cell line was a generous gift from Dr. NaiheJing (Shanghai Institute of Biochemistry and Cell Biology, CAS). Both cell lines were verified by STR matching analysis and tested negative for mycoplasma. Both HEK293 and P19 cell lines were grown in Dulbecco's Modified Eagle Medium (Gibco, C11995500BT) supplemented with 10% fetal bovine serum (FBS) (Biological Industries, 04-001-1A), 100 units/mL penicillin and 100 mg/mL streptomycin (Biological Industries, 03-031-1B). Cell cultures were maintained at 37°C in a humidified incubator with 5% $CO_2$. 0.5 µM RA (Sigma, R2625) was used to induce *Hox* expression in P19 cells.

To achieve gene overexpression or knockdown, HEK293 and P19 cells were transfected by Lipo2000 (Invitrogen, 11668500) according to the manufacturer's instructions. The following siRNAs (RiboBio) were used for *Rnf220* or *Wdr5* knockdown in P19 cells: siG2010160325456075, siG2010160325457167, and siG2010160325458259 were used for *Rnf220* knockdown; siB09924171210, siG131113135429, and siG131113135419 were used for *Wdr5* knockdown.

## Total RNA isolation and qRT-PCR

Tissue and cells were homogenized with 1 mL TRIzol (TIANGEN, DP424), after which 200 µL of chloroform was added to the lysates for phase separation. After centrifugation at 12,000×*g* for 15 min at 4°C, the aqueous phase (500 µL) was transferred to a new tube and mixed with equal volumes of isopropanol for RNA precipitation. After 30 min, RNA pellets were harvested by centrifugation at 12,000×*g* for 15 min at 4°C, twice washed with 75% ethanol, and dissolved in DNase/RNase-free water.

cDNA was synthesized with Strand cDNA Synthesis Kit (Thermo Scientific, K1632) according to the manufacturer's instructions. All reactions were performed at least triplicates with Light Cycler 480 SYBR Green I Master (Roche, 04707516001). Primers used for qRT-PCR are listed in *Table 1*.

## ChIP-qPCR

P19 cells or grinded mouse PN tissues were cross-linked with 1% formaldehyde for 25 min, and stopped with addition of glycine. After gentle centrifuging, samples were collected and lysed with ChIP lysis buffer (150 mM NaCl, 25 mM Tris, pH 7.5, 1% Triton X-100, 0.1% SDS, 0.5% deoxycholate, and complete protease inhibitor cocktail) for 30 min on ice. Genomic DNA was sheared into

**Table 2.** Primers used for ChIP-qPCR.

| Genes | Forward primers | Reverse primers |
|---|---|---|
| *Hoxa1* | GCAGGACAAGGTTGATGGG | GCAGGTGGGAGGGACAGAT |
| *Hoxa3* | GGACAGACTCGGTGGTAAGA | AGTTCATGTTCACGGTTCCTAT |
| *Hoxa9* | GCAGGAAACACTTTGCCAGA | GCCCGAGTTAGGACCCGTA |
| *Hoxa10* | CTCCTTGCCTCCTTCTTCC | CCTGGGTATCTGAGCATCTAA |
| *Hoxb3* | CCGAGGACGGACCGAAGAT | CCCTGAACTGGACCACCAT |
| *Hoxb4* | GAAGAACGCACGGAAAGTAAG | GGGAAAGAATATGAGCGGAGT |
| *Hoxb7* | CCTTAGGGACGCCTTGGTC | ACGCAGGGATTGAATGTTCG |
| *Hoxb8* | GCCATTGAATTTCTATCCCAC | GGTGAGGCAAGCTAAAGCAG |
| *Hoxc6* | CTTCTCCTCTGCCCTCTTC | GTTAGTTAATACATGGACCTCT |
| *Hoxc8* | GTCGTGGATTGATGAACGCG | TCTGCTCACTGTCGGTAGG |
| *Hoxc9* | TGTGCCTTGAGTCACTTTGC | CTTGCTCCACTTCTCCAGAT |
| *Hoxc10* | TTTTCTTTGGGTCCTCGTAAA | AGTCTAGGGAGCCATTTGTC |
| *Hoxd3* | TTTTCCGAGTCCTATTGCTTG | CTGTATCATCTGCCCTCTATC |
| *Hoxd8* | AGGACTTTGATTCGCTTTGATA | CGAGGTTGACGGATTGATTG |
| *Hoxd9* | AACCTACCCTCGGAGATGC | GCACTGGAGTCCCAAGGAG |
| *Hoxd10* | GGAGGGATGTTTCCGAACT | CACATACCCAGGCAGAACG |

200–500 bp fragments by sonication. After centrifugation, the supernatants containing chromatin fragments were collected and immunoprecipitated with anti-IgG (2 µg, ProteinTech, B900610), anti-H3K27me3 (2 µg, Abcam, ab192985, RRID: AB_2650559), or anti-H3K4me3 antibodies (2 µg, Abcam, ab8580, RRID: AB_306649) at 4°C overnight. Then, the immunoprecipitates coupled with protein A/G agarose beads (Santa Cruz, sc-2003) were washed sequentially by a low salt washing buffer, high salt washing buffer, LiCl washing buffer, and TE buffer. The immunoprecipitated chromatin fragments were eluted by 500 µL elution buffer for reversal of cross-linking at 65°C overnight. Input or immunoprecipitated genomic DNA was purified by the QIAquick PCR Purification Kit (QIAGEN, 28104) and used as a template for quantification PCR. The primers we used were listed in *Table 2*.

## In utero microinjection

Pregnant mice were administered isoflurane for deep anesthesia. Following this, a laparotomy was performed to carefully extract the embryos, which were then placed on a sterile surgical drape. WDR5-IN-4 (100 µg, MedChemExpress, HY-111753A), containing 0.05% Malachite Green reagent for tracing, was injected into the hindbrain of E15.5 embryos using a finely tapered borosilicate needle. To minimize the risk of spontaneous abortion, injections were spaced for the selected embryos. Following the injections, the uterus was returned to the abdominal cavity and infused with 2 mL of 37°C, 0.9% saline solution. The peritoneum and abdominal skin were then sutured. Finally, the mice were placed on a heating pad to facilitate recovery from anesthesia.

## Ubiquitination assay, immunoprecipitation, and western blot analysis

In vivo ubiquitination, immunoprecipitation, and western blot assays were carried out as previously described (*Ma et al., 2014*). HEK293 cells were transfected in six-well plates, and at 48 hr post transfection cells were lysed in 500 µL of lysis buffer (50 mM Tris-HCl [pH 7.4], 150 mM NaCl, 5 mM EDTA [pH 8.0], and 1% Triton X-100) that contained a protease inhibitor mixture (Roche Applied Science) for 30 min on ice; mouse tissues were fully grinded and lysed with RIPA Strong Lysis (Beyotime, P0013B) containing a protease inhibitor mixture for 30 min on ice. Following centrifugation at 12,000 rpm for 10 min at 4°C, 50 µL supernatant was mixed with loading buffer and incubate boiled at 95°C for 5 min, the remaining supernatant was coated with antibody and A/G agarose beads (or FLAG beads) overnight. After washing five times, the bound proteins were eluted with SDS loading buffer at 95°C for 5 min. Final total lysates and immunoprecipitates were subjected to SDS-PAGE and western blot analysis. The following primary antibodies were used for botting: anti-RNF220 (Sigma-Aldrich, HPA027578, RRID: AB_10601482), anti-WDR5 (D9E11) (Cell Signaling Technology, 13105, RRID: AB_2620133), anti-RING1B (D22F2) (Cell Signaling Technology, 5694S, RRID: AB_10705604), anti-SIN3B (AK-12) (Santa Cruz, sc-768, RRID: AB_2187787), anti-EZH2 (ProteinTech, 21800-1-AP, RRID: AB_10858790), anti-SUZ12 (Bethyl, A302-407A, RRID: AB_1907290), anti-RYBP (A-1) (Santa Cruz, sc-374235, RRID: AB_10989572), anti-CBX6 (H-1) (Santa Cruz, sc-393040, RRID: AB_2923357), anti-CBX7 (G-3) (Santa Cruz; sc-376274, RRID: AB_10989202), anti-CBX8 (C-3) (Santa Cruz, sc-374332, RRID: AB_10990104), anti-PHC1 (D-10) (Santa Cruz, sc-390880), anti-α-Tubulin (ProteinTech, 66031-1-Ig, RRID: AB_11042766), anti-FLAG (Sigma-Aldrich; F-7425, RRID: AB_439687), anti-myc (ProteinTech; 16286-1-AP, RRID: AB_1182162), and anti-Ub (P4D1) (Santa Cruz, sc-2007, RRID: AB_631740).

## snRNA-seq library preparation, sequencing, and data analysis

snRNA-seq libraries were prepared using the Split-seq platform (*Butler et al., 2018*; *Rosenberg et al., 2018*). Freshly harvested mouse pons tissues underwent nuclear extraction following previously described protocols (*Butler et al., 2018*; *Rosenberg et al., 2018*). In brief, mouse brain tissues were transferred into a 2 mL Dounce homogenizer containing 1 mL homogenizing buffer (250 mM sucrose, 25 mM KCl, 5 mM MgCl$_2$, and 10 mM Tris-HCl [pH = 8.0], 1 mM DTT, RNase Inhibitor, and 0.1% Triton X-100). The mixture was subjected to five strokes with a loose pestle, followed by 10 strokes with a tight pestle. The resulting homogenates were filtered through a 40 µm strainer into 5 mL Eppendorf tubes and subsequently centrifuged for 4 min at 600×*g* and 4°C. The pellet was re-suspended and washed in 1 mL of PBS containing RNase inhibitors and 0.1% BSA. The nuclei were again filtered through a 40 µm strainer and quantified. These nuclei were partitioned into 48 wells, each of which contained a barcoded, well-specified reverse transcription primer, to enable in-nuclear reverse transcription. Subsequent second and third barcoding steps were carried out through ligation

reactions. After the third round of barcoding, the nuclei were divided into 16 aliquots and lysed prior to cDNA purification. The resulting purified cDNA was subjected to template switching and qRT-PCR amplification, which was halted at the beginning of the plateau stage. Finally, the purified PCR products (600 pg) were used to generate Illumina-compatible sequencing libraries. Distinct, indexed PCR primer pairs were employed to label each library, serving as the fourth barcode.

The libraries underwent sequencing on the NextSeq system (Illumina) using 150-nucleotide kits and paired-end sequencing protocols. In this arrangement, Read 1 covered the transcript sequences and Read 2 contained the UMI and UBC barcode combinations. Initially, a six-nucleotide sequencing index, serving as the fourth barcode, was appended to the ends of Read 2. Subsequently, reads with more than one mismatching base in the third barcode were excluded from the dataset. Furthermore, reads containing more than one low-quality base (Phred score≤10) within the UMI region were also discarded. The sequencing results were aligned to exons and introns in the reference genome (Genome assembly GRCm38) and aggregated intron and exon counts at the gene level were calculated by kallisto and bustools software as described (https://bustools.github.io/BUS_notebooks_R).

Following export of the matrix, quality control measures were performed to remove low-quality cells and potential doublets, as described previously (*Rosenberg et al., 2018*). After filtering, a total of 125,956 and 122,266 cells for WT and *Rnf220*[+/-] pons respectively remained for subsequent analysis. Seurat v2 (*Butler et al., 2018*.) was used to identify HVGs within each group. Principal component analysis and UMAP were performed to embed each cell from the reduced principal component space on a 2D map. Then clustering of cell populations and identification of differentially expressed genes (DEGs) were carried out as previously described (*Ma and Mao, 2022b*). We annotated the embryonic cell populations and lineages based on their DEGs (*Supplementary file 3*).

## Mouse brain stereotactic injection and neuronal tracing

Mouse brain stereotactic injection and neuronal tracing were carried out as previously described with minor modifications (*Xu et al., 2021*). Adult mice (2–3 months of age) were used for anterograde monosynaptic neuronal tracing. The mice were first deeply anesthetized with isoflurane and placed into a stereotactic apparatus with the front teeth latched onto the anterior clamp. The mouth and nose of the mice were covered with a mask to provide isoflurane and keep them in an anesthetic state during the operation. The head was adjusted and maintained in horizontal by inserting ear bars into the ear canal. The hair of the head was shaved with an electric razor and cleaned using a cotton swab which was dipped in 75% alcohol. Cut the scalp along the midline with surgical scissors and make sure the bregma, lambda, and the interaural line were exposed. The intersection between the lambda and interaural line was set to zero. The coordinates ±1.75, −4.9, −1.65 were applied for motor cortex localization. Each mouse received a single injection of 500 nL of rAAV-hSyn-EGFP-WPRE-hGH-poly A (Brainvta, PT-1990) viral fluid in the hemisphere. After the injection, mice were bred for another 3 weeks for neuronal tracing before their brains were harvested for analysis.

## Slice preparation

Mice subjected to stereotactic injection were euthanized, and whole brains were collected for the preparation of frozen sections. Following fixation in 4% paraformaldehyde (diluted in PBS) for 48 hr, the brains were successively treated with 20% sucrose (diluted in PBS) for 48 hr and 30% sucrose for 24 hr. Brain tissues were then embedded in optimal cutting temperature compound (SAKURA, 4583) at −20°C and sliced along the sagittal plane at a thickness of 40 μm.

## Quantification and statistical analysis

PN and fluorescence length and area were statistically analyzed by ImageJ software (National Institute of Health). Statistical analyses were performed using GraphPad Prism software (GraphPad Software Inc, La Jolla, CA, USA). All experiments were repeated at least two times. Comparisons were performed using the two-tailed Student's t-test. p-Values of less than 0.05 were considered statistically significant (*), 0.01 were considered statistically significant (**), and 0.001 were considered statistically significant (***).

## Acknowledgements

We are grateful to all members of the Mao and Zhang laboratories for discussion of and comments on the manuscript. We would like to thank the Core Technology Facility of Kunming Institute of Zoology (KIZ), Chinese Academy of Sciences (CAS) for providing us with service.

## Additional information

### Funding

| Funder | Grant reference number | Author |
| --- | --- | --- |
| National Key R&D Program of China | 2021YFF0702700 | Bingyu Mao |
| Yunnan Fundamental Research Projects | 202101AU070137 | Huishan Wang |
| National Natural Science Foundation of China | 8240055266 | Huishan Wang |
| National Natural Science Foundation of China | 32170965 | Pengcheng Ma |
| Yunnan Fundamental Research Projects | 202205AC160065 | Pengcheng Ma |
| Yunnan Fundamental Research Projects | 202201AW070009 | Pengcheng Ma |
| Yunnan Fundamental Research Projects | 202301AS070059 | Pengcheng Ma |
| Yunnan Fundamental Research Projects | 202401AT070187 | Huishan Wang |

The funders had no role in study design, data collection and interpretation, or the decision to submit the work for publication.

### Author contributions

Huishan Wang, Data curation, Formal analysis, Funding acquisition, Validation, Investigation, Visualization, Methodology, Writing – original draft; Xingyan Liu, Data curation, Software, Formal analysis, Visualization, Methodology; Yamin Liu, Validation, Investigation, Visualization; Chencheng Yang, Investigation, Visualization, Methodology; Yaxin Ye, Investigation, Methodology; Xiaomei Yu, Visualization; Nengyin Sheng, Conceptualization; Shihua Zhang, Data curation, Software, Formal analysis; Bingyu Mao, Pengcheng Ma, Conceptualization, Resources, Formal analysis, Supervision, Funding acquisition, Visualization, Project administration, Writing – review and editing

### Author ORCIDs

Huishan Wang (iD) https://orcid.org/0000-0001-5755-6606
Pengcheng Ma (iD) https://orcid.org/0000-0002-1067-8021

### Ethics

This study was performed in strict accordance with the recommendations in the Animal Care and Use Committee of the Kunming Institute of Zoology, Chinese Academy of Sciences. All of the animals were handled according to Animal Care and Use Committee of the Kunming Institute of Zoology, Chinese Academy of Sciences (Permit Number: IACUC-PA-2021-07-018). All surgery was performed under sodium pentobarbital or isoflurane, and every effort was made to minimize suffering.

Reviewer #1 (Public review): https://doi.org/10.7554/eLife.94657.3.sa1
Author response https://doi.org/10.7554/eLife.94657.3.sa2

# Additional files

## Supplementary files

• Supplementary file 1. Differently expressed genes identified using microarray between wild-type (WT) and *Rnf220*<sup>+/-</sup> mice. Whole-mount brain from E18.5 mice were used (n=2 in WT group and n=3 in *Rnf220*<sup>+/-</sup> group).

• Supplementary file 2. Differently expressed genes identified using microarray between wild-type (WT) and *Rnf220*<sup>-/-</sup> mice. Whole-mount brain from E18.5 mice were used (n=2 in WT group and n=3 in *Rnf220*<sup>-/-</sup> group).

• Supplementary file 3. Uniquely and highly expressed genes of each cluster in single-nucleus RNA sequencing (snRNA-seq). The pons from 2 months' mice were used (n=3 mice per group).

• MDAR checklist

## Data availability

All the snRNA-seq and RNA-seq raw data have been deposited in the GSA (https://ngdc.cncb.ac.cn/gsa/) with accession number is CRA013111.

The following dataset was generated:

| Author(s) | Year | Dataset title | Dataset URL | Database and Identifier |
|---|---|---|---|---|
| Huishan W | 2024 | Hindbrain pattern maintenance in mouse | https://ngdc.cncb.ac.cn/gsa/browse/CRA013111 | Genome Sequence Archive, CRA013111 |

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
