## [Editor Report · eLife Assessment]

This **valuable** study focuses on gene regulatory mechanisms essential for hindbrain development. Through molecular genetics and biochemistry, the authors propose a new mechanism for the control of Hox genes, which encode highly conserved transcription factors essential for hindbrain development. The strength of evidence is **solid**, as most claims are supported by the data. This work will be of interest to developmental biologists.

---

## [Referee Report · Reviewer #1 (Public review)]

The manuscript by Wang et al. investigates the role of Rnf220 in hindbrain development and Hox expression. The authors suggest that Rnf220 controls Hox expression in the hindbrain through regulating WDR5 levels. The authors combine in vivo experiments with experiments in P19 cells to demonstrate this mechanism. However, the in vivo data does not provide strong support for the claims the authors make and the role of Rnf in Hox maintenance and pons development is unclear.

While the authors partially addressed some of the issues raised in the first round of reviews, and the in vitro data showing a relationship between Rnf220 and WDR5 is convincing, some issues still remain about the experimental evidence supporting their claims and the relationship of this work with previous studies demonstrating the role of Hox proteins in pontine nuclei in vivo.

The authors say they were unable to detect Hox levels via in situ hybridization at late embryonic stages, stating that the levels are likely too low to be detected-yet they are presumably high enough to cause ectopic targeting of pontine neurons. Work from the Rijli group, which the authors cite, shows that Hox3-5 paralogs can be clearly detected both by in situ and by staining with commercially available antibodies. Since a major claim of this paper is the upregulation of Hox genes in Rnf220+/- mice through WDR5 regulation, the authors need to show this more convincingly. The inability to detect Hox upregulation, and subsequent rescue, by means other than qPCR in vivo remains a major weakness of the paper. The authors also do not discuss how broad upregulation of all Hox paralogs leads to the changes in PN targeting in the context of previous work.

The links between Wdr5 expression, epigenetic modifications, Hox expression and axon mistargeting in vivo remains somewhat tenuous. For example, the authors show epigenetic modification changes in some Hox genes, but not Hox5 paralogs, and only show the rescue by Wdr5 KO in vitro. Similarly, they do not attempt to show rescue of axon targeting in vivo after presumably restoring Hox levels by Wdr5 inhibition or knockdown.

---

## [Author Response]

The following is the authors’ response to the original reviews.

**Reviewer 1:**
(1) A major issue throughout the paper is that Hox expression analysis is done exclusively through quantitative PCR, with values ranging from 2-fold to several thousand-fold upregulation, with no antibody validation for any Hox protein (presumably they are all upregulated).

Thank you for your comment.

We tried to verify the stimulated *Hox* expression pattern by in situ hybridization. Although in early embryos (E9.5) we could detect clearly hox (*i.e. Hox8 and Hox9* in Author response image 1) expression patterns in the neural tube by whole mount in situ hybridization, we failed to detect a clear pattern in the brain stem at E18.5 either in whole mount tissue or on sections. That’s one reason that we turned to single nuclear RNA-seq instead.

This is likely due to their low expression levels at late developmental stages and need to be detected by more sensitive method. However, we estimated that the stimulated expression levels of the representative *Hox* genes are at least comparable to the physiological levels at posterior spinal cord to evoke a functional effect.

**Author response image 1. sa2fig1:** Some *Hox8 and Hox9* expression pattern in E9. 5 embryos.

(2) In Figure 1, massive upregulation of most Hox genes in the brainstem is shown after e16.5 but the paper quickly focuses on analysis of PN nuclei. What are the other consequences of this broad upregulation of Hox genes in the brainstem? There is no discussion of the overall phenotype of the mice, the structure of the brainstem, the migration of neurons, etc. The very narrow focus on motor cortex projections to PN nuclei seems bizarre without broad characterization of the mice, and the brainstem in particular. There is only a mention of "severe motor deficits" from previous studies, but given the broad expression of Rnf220, the fact that is a global knockout, and the effects on spinal cord populations shown previously the justification for focusing on PN nuclei does not seem strong.

Thank you for your comment.

Although RNF220 is important for the dorsal-ventral patterning of the spinal cord as well as the hindbrain during embryonic development, the earlier neural patterning and differentiation are normal in the *Rnf220+/-* mice (Wang *et al*., 2022). However, these mice showed reduced survival and motility to various degree postnatally (Ma *et al*., 2019; Ma *et al*., 2021), likely suggesting a dosage dependent role of RNF220 in maintaining late neural development. As our microarray assay showed the deregulation of the *Hox* genes in the brain, we followed this direction in this study and narrowed down the affected region to the pons. Our single nuclear RNA-Seq (snRNA-seq) data further shows that the *Hox* de-regulation mainly occurred in 3 clusters of neurons. However, the pons is complex and contains tens of nuclei. And the current resolution of our data does not support to assign a clear identity to each of them. Although it is clear that more nuclei are likely affected, the PN (cluster7) is the only cluster we can identify to follow in the current study.

As to general effect of RNF220 haploinsufficiency on the brainstem, we carried out Nissl staining assays and found no clear difference in neuronal cell organization between WT and *Rnf220+/-* pons (revised Figure 2-figure supplement 2).

(3) It is stated that cluster 7 in scRNA-seq corresponds to the PN nuclei. The modest effect shown on Hox3-5 expression in that data in Figure 1 is inconsistent with the larger effect shown in Figure 2.

Thank you for your comment.

Due to the low efficiency of snRNA-seq and the depth of the sequencing, the quantification of the *Hox* expression based on the snRNA-seq data is likely less accurate as the qRT-PCR. In addition, only mRNAs in the nuclear could be captured by snRNA-seq, while mRNAs in both the nuclear and cytoplasm were reversed-transcribed and examined for qRT-PCR assays in Figure 2A.

(4) Presumably, Hox genes are not the only targets of Rnf220 as shown in the microarray/RNA-sequencing data. There is no definitive evidence that any phenotypes observed (which are also not clear) are specifically due to Hox upregulation. The only assay the authors use to look at a Hox-dependent phenotype in the brainstem is the targeting of PN nuclei by motor cortex axons. This is only done in 2 animals and there are no details as to how the data was analyzed and quantified. The only 2 images shown are not convincing of a strong phenotype, they could be taken at slightly different levels or angles. At the very least, serial sections should be shown and the experiment repeated in more animals. There is also no discussion of how these phenotypes, if real, would relate to previous work by the Rijli group which showed very precise mechanisms of synaptic specificity in this system.

Thank you for your comments and suggestions.

The deregulation of *Hox* is the most obvious phenomena observed from the RNA-seq data, and we tried to assign its specific phenotypic effect in this study. As the roles of *Hox* in PN patterning and circuit formation is well established, we focused on the PN in the following study. Based on literature, we carried out the circuit analysis to examine the targeting of PN neurons by the motor cortex axons. A cohort of additional animals with different genotypes (n=10 for WT and n=9 for *Rnf220+/-*) were used to repeat the experiment and we got the same conclusion. More detailed information on data analysis and serial images were included in the revised manuscript and figure legends.

(5) The temporal aspect of this regulation in vivo is not clear. The authors show some expression changes begin at e16.5 but are also present at 2 months. Is the presumed effect on neural circuits a result of developmental upregulation at late embryonic stages or does the continuous overexpression in adult mice have additional influence? Are any of the Hox genes upregulated normally expressed in the brainstem, or PN specifically, at 2 months? Why perform single-cell sequencing experiments at 2 months if this is thought to be mostly a developmental effect? Similarly, the significance of the upregulated WRD5 in the pons and pontine nuclei at 2 months in Figure 3 is not clear.

Thank you for your comment.

The spatial and temporal expression pattern of *Hox* genes is established at early embryonic stages and then maintained throughout developmental stage in mammals. As we have shown, the de-repression of *Hox* genes is a long-lasting defect in *Rnf220+/-* mice beginning at late embryonic stages. Since the neuronal circuit is established after birth in mice, we speculated that the neuronal circuit defects from motor cortex to PN neurons were due to the long-lasting up-regulation of *Hox* genes in PN neurons. We could not distinguish the effect on neural circuit a result of *Hox* genes developmental upregulation or continuous overexpression in adult mice. An inducible knockout mouse model may help to answer this question in the future. The discussion on this point was included in the revised manuscript.

We carried out snRNA-seq analysis using pons tissues from adult mice aiming to identify the specific cell population with *Hox* up-regulation, which we failed to specify by in situ hybridization.

We repeated the related experiments in the original Figure 3 and some of the blot images were replaced and quantified.

(6) In Figure 3C, the levels of RNF220 in wt and het don't seem to be that different.

We repeated the experiments and changed the related image in the revised Figure 3C.

(7) Based on the single-cell experiments, and the PN nuclei focus, the rescue experiments are confusing. If the Rnf220 deletion has a sustained effect for up to 2 months, why do the injections in utero? If the focus is the PN nuclei why look at Hox9 expression and not Hox3-5 which are the only Hox genes upregulated in PN based on sc-sequencing? No rescue of behavior or any phenotype other than Hox expression by qPCR is shown and it is unclear whether upregulation of Hox9 paralogs leads to any defects in the first place. The switch to the Nes-cre driver is not explained. Also, it seems that wdr5 mRNA levels are not so relevant and protein levels should be shown instead (same for rescue experiments in P19 cells).

Thank you for your comments.

Since our data suggest that the upregulation of *Hox* genes expression is a long-lasting effect beginning at the late embryonic stage of E16.5, we conducted the rescue experiments by in utero injection of WDR5 inhibitor at E15.5 and examined the expression of *Hox* genes at E18.5. Although it is also necessary to examine whether the rescue effect by WDR5 inhibitor injection is also a long-lasting effect at adult stages, it is difficult to distinguish the embryos or pups when they were given birth. As a supplement, rescue assays with genetic ablation of *Wdr5* gene were conducted and the results showed that genetic ablation of a single copy of *Wdr5* allele could revere the upregulation of *Hox* genes by RNF220 haploinsufficiency in the hindbrains at P15.

Most of the upregulated *Hox* genes including both *Hox9* and *Hox3-5* were examined in our rescue experiments. Since this study focuses on the PN nuclei, the results of *Hox3-5* genes were shown in the revised main Figure 6.

We conducted rescue experiments by deleting *Wdr5* in neural tissue using _Nestin-Cr_e mice because *Wdr5+/-* mice is embryonic lethal. And the up-regulation of *Hox* genes could be also observed in the hindbrains of *Rnf220fl/wt*; *Nestin-Cre* mice. Although *Rnf220fl/wt; Wdr5fl/wt; Nestin-Cre* mice are viable and could survive to adult stages, developmental defects in the forebrains, including cerebral cortex and hippocampus, were observed in *Rnf220fl/wt;Wdr5fl/wt;Nestin-Cre* mice. Therefore, no rescue of behavior tests was conducted in this study. We believe that it is out of the scope of this study to discuss the role of WDR5 in the development of forebrains.

The potential defects due to the up-regulation of *Hox9* paralogs awaits further investigations.

*Wdr5* mRNA levels were firstly examined to confirm the genetic deletion or siRNA mediated knockdown of *Wdr5* genes. We have carried out western blot to examine the WDR5 protein levels and the results were included in the revised Figure 3.

(8) What is the relationship between Retinoic acid and WRD5? In Figure 3E there is no change in WRD5 levels without RA treatment in Rnf KO but an increase in expression with RA treatment and Rnf KO. However, the levels of WRD5 do not seem to change with RA treatment alone. Does Rnf220 only mediate WDR5 degradation in the presence of RA? This does not seem to be the case in experiments in 293 cells in Figure 4.

Thank you for your comment.

We believe that the regulation of WDR5 and *Hox* expression by RNF220 is context dependent and precisely controlled in vivo, depending on the molecular and epigenetic status of the cell, which is fulfilled by RA treatment in P19 cells. In Figure 4, the experiment is based on exogenous overexpression assays, which might not fully reflect the situation in vivo.

(9) Why are the levels of Hox upregulation after RA treatment so different in Figure 5 and Figure Supplement 5?

In Figure.5C, the *Hox* expression levels were normalized against the control group in the presence of RA; while in Figure Supplement 5 they were normalized to the control group without RA treatment.

(10) In Figures 4B+C which lanes are input and which are IP? There is no quantitation of Figure 4D, from the blot it does look that there is a reduction in the last 2 columns as well. The band in the WT flag lane seems to have a bubble. Need to quantitate band intensities. Same for E, the effect does not seem to be completely reversed with MG132.

Thanks for pointing this out. The labels were included in the revised Figure 4B and 4C.

We repeated the experiments for Figure 4D and 4E. Some of bot images were replaced and quantified in the revised Figure 4D and 4E.

**Reviewer 2:**
(1) Figure 1E shows that Rnf220 knockdown alone could not induce an increase in Hox expression without RA, which indicates that Rnf220 might endogenously upregulate Retinoic acid signaling. The authors should test if RA signaling is downstream of Rnf220 by looking at differences in the expression of Retinaldehyde dehydrogenase genes (as a proxy for RA synthesis) upon Rnf220 knockdown.

Thank you for your comment and suggestion.

Two sequential reactions are required for RA synthesis from retinol, which catalyzed by alcohol dehydrogenases (ADHs)/ retinol dehydrogenase (RDH) and retinaldehyde dehydrogenase (RALDHs also known as ALDHs) respectively. When RA is no longer needed, it is catabolized by cytochrome enzymes (CYP26 enzymes) (Niederreither, *et al*.,2008; Kedishvili *et al.*, 2016). Here, we test ADHs、ALDHs and CYP26 enzymes in E16.5 WT and *Rnf220-/-* embryos.

The results are as follows. ADH7 and ADH10 are slightly upregulated. ALDH1 and ALDH3 are upregulated and downregulated in *Rnf220-/-* embryos, respectively, but there is no significant change in the expression of ALDH2, which plays a key role in RA synthesis during embryonic development (Niederreither, *et al*.,2008). Furthermore, Cyp26a1 which responsible for RA catabolism was upregulated in *Rnf220-/-* embryos. Collectively, these data do not support a clear effect on RA signaling by RNF220.

**Author response image 2. sa2fig2:** The effect of *Rnf220* on RA synthesis and degradation pathways.

(2) In Figure 2C-D further explanation is required to describe what criteria were used to segment the tissue into Rostral, middle, and caudal regions. Additionally, it is unclear whether the observed change in axonal projection pattern is caused due to physical deformation and rearrangement of the entire Pons tissue or due to disruption of Hox3-5 expression levels. Labeling of the tissue with DAPI or brightfield image to show the structural differences and similarities between the brain regions of WT and Rnf220 +/- will be helpful.

Thank you for your comment and suggestion.

More information on the quantification of the results shown in Figure 2C-D was included in our revised manuscript. We carried out Nissl staining assays using coronal sections of the brainstem and found that there is no significant difference in neuronal cell organization between WT and *Rnf220+/-* (revised Figure 2-figure supplement 2).

(3) Line 192-195. These roles of PcG and trxG complexes are inconsistent with their initial descriptions in the text - lines 73-74.

We are sorry for the mistake. We carefully revised the related descriptions to avoid such mistake. Thank you.

(4) In Figure 4D, the band in the gel seems unclear and erased. Please provide a different one. These data show that neither Rnf220 nor wdr5 directly regulates Hox gene expressions. The effect of double knockdown in the presence of RA suggests that they work together to suppress Hox gene expression via a different downstream target. This point should be addressed in the text and discussion section of the paper. example for the same data which shows a full band with lower intensity.

Thank you for your suggestion.

We repeated the experiment of Figure 4D and some of the blot images were replaced in the revised Figure 4D.

Indeed, in the presence of RA, knockdown of *Rnf220* alone can upregulate the expression *Hox* genes (Figure 5C). Knockdown of *Wdr5* could reverse the upregulation of *Hox* genes in RNF220 knockdown cells, suggesting that *Rnf220* regulated *Hox* gene expression in a *Wdr5* dependent manner. However, in the absence of RA, none of *Rnf220* knockdown, *Wdr5* knockdown or *Rnf220* and *Wdr5* double knockdown had a significant effect on the expression of *Hox* genes in P19 cells. It seems that RA signaling plays a crucial role for the regulation of RNF220 to WDR5 in P19 cells and discussion on this point was included in the revised manuscript.

(5) In Figure 4G the authors could provide some form of quantitation for changes in ubiquitination levels to make it easier for the reader. They should also describe the experimental procedures and conditions used for each of the pull-down and ubiquitination assays in greater detail in the methods section.

Thank you for your suggestion.

The quantitation and statistics for the original Figure 4G were included in the revised Figure 4. More information on the biochemical assays was included in the “Methods and Materials” section of our revised manuscript.

(6) Figure 5 shows that neither Rnf220 nor wdr5 directly regulate Hox gene expressions. The effect of double knockdown in the presence of RA suggests that they work together to suppress Hox gene expression via a different downstream target.

Thank you for your comment.

In fact, knockdown of *Rnf220* alone can upregulate the expression *Hox* genes in the presence of RA (Figure 5C). Furthermore, knockdown of *Wdr5* could reverse the upregulation of *Hox* genes in *Rnf220* knockdown cells, which suggest that *Rnf220* regulated *Hox* gene expression in a *Wdr5* dependent manner. However, in the absence of RA, none of *Rnf220* knockdown, *Wdr5* knockdown or *Rnf220* and *Wdr5* double knockdown had a significant effect on the expression of *Hox* genes in P19 cells. It seems that RA signaling plays a crucial role for the regulation of RNF220 to WDR5 in P19 cells and discussion on this point was included in the revised manuscript.

(7) In Figure 6, while the reversal of changes in Hox gene expression upon concurrent Rnf220; Wdr5 inhibition highlights the importance of Wdr5 in this regulatory process, the mechanistic role of wdr5 and its functional consequences are unclear. To answer these questions, the authors need to: (i) Assay for activated and repressive epigenetic modifications upon double knockdown of Rnf220 and Wdr5 similar to that shown in Figure 3- supplement 1. This will reveal if wdr5 functions according to its intended role as part of the TrxG complex. (ii) The authors need to assay for changes in axon projection patterns in the double knockdown condition to see if Wdr5 inhibition rescues the neural circuit defects in Rnf220 +/- mice.

Thank you for your suggestion.

Although it is also necessary to examine whether the rescue effect by WDR5 inhibitor injection in uetro is also a long-lasting effect for neuronal cirtuit at adult stages, it is difficult to distinguish the embryos or pups when they were given birth. Although *Rnf220fl/wt;Wdr5fl/wt;Nestin-Cre* mice are viable and could survive to adult stages, developmental defects in the forebrains, including cerebral cortex and hippocampus, were observed in *Rnf220fl/wt;Wdr5fl/wt;Nestin-Cre* mice. Therefore, no rescue effect on defects of behavior and neuronal circuit were examined in this study. Maybe, a PN nuclei specific inducible Cre mouse line could help toward this direction in the future.

We carried out ChIP-qPCR and tested activated and repressive epigenetic modifications upon double knockdown of *Rnf220* and *Wdr5* in P19 cell line and found *Rnf220* and *Wdr5* double knockdown recured Hox epigenetic modification to a certain degree (Figure 6-figure supplement 1).

References

Kedishvili, N.Y. 2016. Retinoic acid synthesis and degradation. *Subcell Biochem*, 81:127-161. DOI: 10.1007/978-94-024-0945-1_5, PMID: 2783050

Ma, P., Li, Y., Wang, H., Mao, B., Luo, Z.-G. 2021. Haploinsufficiency of the TDP43 ubiquitin E3 ligase RNF220 leads to ALS-like motor neuron defects in the mouse. *Journal of Molecular Cell Biology, 13*: 374-382. DOI: 10.1093/jmcb/mjaa072, PMID: 33386850

Ma, P., Song, N.-N., Li, Y., Zhang, Q., Zhang, L., Zhang, L., Kong, Q., Ma, L., Yang, X., Ren, B., Li, C., Zhao, X., Li, Y., Xu, Y., Gao, X., Ding, Y.-Q., Mao, B. 2019. Fine-Tuning of Shh/Gli Signaling Gradient by Non-proteolytic Ubiquitination during Neural Patterning. *Cell Rep, 28*: 541-553.e544. DOI: 10.1016/j.celrep.2019.06.017, PMID: 31291587

Niederreither, K., Dollé, P. 2008. Retinoic acid in development: towards an integrated view. *Nat Rev Genet*, 9: 541-53. DOI: 10.1038/nrg2340, PMID: 18542081

Wang, Y.-B., Song, N.-N., Zhang, L., Ma, P., Chen, J.-Y., Huang, Y., Hu, L., Mao, B., Ding, Y.-Q. 2022. Rnf220 is Implicated in the Dorsoventral Patterning of the Hindbrain Neural Tube in Mice. *Front Cell Dev Biol, 10*. DOI: 10.3389/fcell.2022.831365, PMID: 35399523